# Operator-Based Generalization Bounds for Multitask Deep Learning

## Abstract

We study generalization bounds for compositions of functions through an operator-theoretic (Koopman) framework. Existing analyses in this direction are primarily restricted to scalar-valued settings and to Sobolev-type reproducing kernel Hilbert spaces (RKHSs), where the resulting bounds depend on smoothness parameters. We extend this framework to vector-valued RKHSs, enabling the analysis of multi-output function classes and making explicit how task-coupling kernels enter the resulting Rademacher complexity bounds. Within this setting, we derive bounds that depend on operator norms, singular values, and determinant-based geometric quantities associated with the underlying linear maps. We further investigate a vector-valued Brownian/Cameron–Martin formulation based on the one-dimensional Brownian kernel and its associated Cameron–Martin space. This provides an alternative function-space perspective to the Sobolev framework and yields corresponding Koopman-based complexity bounds for compositions of invertible transformations in the Brownian/Cameron–Martin setting. The resulting estimates exhibit a different dependence on the layer transformations and activation operators, reflecting the distinct geometry of the underlying function space. We additionally study a shared operator-learning formulation for multitask transfer in vector-valued RKHSs, deriving an exact representer theorem, a finite-dimensional reduction of the corresponding operator-learning problem, and transfer bounds for the induced operator class. Finally, we provide empirical studies on synthetic data and MNIST that investigate the behavior of the resulting Sobolev-based and Brownian-inspired complexity factors during training.

## 1 Introduction

Understanding the generalization properties of deep networks remains a central challenge in learning theory. Classical approaches control capacity either through parameter counts or through norm-based quantities Bartlett & Mendelson (2002); Mohri et al. (2018). While norm-based bounds avoid explicit dependence on width, they often exhibit unfavorable scaling with depth and capture only coarse geometric information about the weight matrices (Neyshabur et al., 2015; Golowich et al., 2020; Bartlett et al., 2017; Wei & Ma, 2019; Liu et al., 2024). Compression-based approaches account for low-rank structure Arora et al. (2018), but do not address generalization in full-rank regimes.

A complementary perspective models neural network layers as Koopman operators acting on function spaces. This operator-theoretic viewpoint allows generalization bounds to be expressed in terms of spectral properties of the underlying transformations, and has been shown to yield improved behavior in scalar-valued Sobolev RKHS settings Hashimoto et al. (2024). However, existing analyses are restricted in two important ways: they are formulated for scalar outputs, and they rely on Sobolev-type RKHSs with smoothness constraints $s > d/2$, which impose structural limitations on the admissible function classes.

**A function-space perspective.** Rather than refining existing bounds within a fixed functional setting, this paper adopts a different viewpoint: we study how *changing the underlying function space* affects operator-based generalization bounds. In particular, we extend the Koopman framework to vector-valued RKHSs

(vvRKHSs), and investigate a Brownian/Cameron–Martin function-space formulation that differs qualitatively from the Sobolev setting. This shift leads to a different operator-theoretic structure in the resulting complexity bounds.

**Overview of results.** Working in vvRKHSs, we derive Koopman-based Rademacher complexity bounds for multi-output function classes. These bounds express capacity in terms of operator norms, singular values, and determinant-based geometric quantities associated with the weight matrices. We further investigate a Brownian/Cameron–Martin function-space formulation and derive corresponding Koopman-based complexity bounds for invertible architectures. The resulting estimates reflect the distinct geometry of the underlying function space and provide an alternative perspective to the Sobolev-based analysis.

**Our contributions.**  We summarize our contributions as follows:

- **Koopman-based bounds in vector-valued RKHSs.** We extend the Koopman operator framework from scalar-valued to vector-valued RKHSs, deriving Rademacher complexity bounds for multi-output function classes and making explicit the role of task-coupling kernels through determinant and trace terms.

- **Brownian RKHS formulation.** We introduce a vector-valued Brownian integral RKHS setting in which Koopman-based bounds no longer depend on Sobolev smoothness exponents, but instead involve only operator norms and determinant-based geometric quantities.

- **Comparison of functional regimes.** We show that the Brownian formulation yields milder spectral dependence than Sobolev-based bounds in well-conditioned regimes, highlighting a structural difference between the two function-space settings.

- **Shared operator learning and transfer.** We introduce a shared operator-learning formulation for multitask transfer in vector-valued RKHSs, deriving an exact representer theorem, a finite-dimensional reduction of the corresponding operator-learning problem, and transfer bounds for the induced operator class. In this framework, the transferable object is a shared operator acting between Hilbert function spaces, rather than merely an output kernel or a finite-dimensional latent representation.

- **Empirical evaluation.** Experiments on synthetic data and MNIST illustrate the comparative behavior of Sobolev and Brownian bounds during training.

## 2   Related Works

**Norm-based generalization bounds.**  A large body of work studies generalization in deep networks through norm-based capacity measures on the weight matrices (Neyshabur et al., 2015; Golowich et al., 2020; Bartlett et al., 2017). These bounds are typically dependent on spectral or Frobenius norms and avoid explicit dependence on layer width. However they often exhibit unfavorable scaling with depth and capture only coarse geometric information about the transformations. Variants based on reference matrices or margin-based quantities have also been proposed (Wei & Ma, 2020; Ju et al., 2022), while other approaches focus primarily on spectral norm control (Li et al., 2021). In contrast, operator-based analyses characterize generalization through the action of layers on function spaces, allowing bounds to depend on finer spectral quantities such as singular values and determinant-based volume distortions.

**Operator-theoretic (Koopman) approaches.**  Recent work has introduced Koopman operator formulations for analyzing generalization in deep networks (Hashimoto et al., 2024). In this framework, each layer is viewed as inducing a composition operator acting on a reproducing kernel Hilbert space (RKHS), and generalization bounds are expressed in terms of operator norms. Existing results are primarily developed for scalar-valued functions in Sobolev-type RKHSs, where the resulting bounds depend on smoothness parameters and associated spectral quantities. The present work builds on this perspective by extending the analysis to vector-valued RKHSs and by investigating Brownian/Cameron–Martin function-space (van der Vaart & van Zanten, 2008) formulations alongside the Sobolev setting.

**Vector-valued and multi-output settings.** Vector-valued RKHSs provide a natural framework for modeling multi-output function classes through matrix-valued kernels (Evgeniou et al., 2005; Micchelli & Pontil, 2005; Caponnetto et al., 2008; Argyriou et al., 2006; 2008). In statistical learning theory, they have been used to study structured prediction, multitask learning, and kernel methods with coupled outputs. Classical generalization results include Rademacher complexity bounds for linear models (Maurer, 2006), excess risk bounds under trace-norm regularization (Pontil & Maurer, 2013), multitask representation-learning guarantees (Maurer et al., 2016), and local complexity analyses (Yousefi et al., 2018). While these works characterize how task coupling affects capacity, they do not address operator-based formulations of deep compositions.

This paper combines these two directions by developing operator-based generalization bounds in vector-valued RKHSs and by investigating a Brownian/Cameron–Martin function-space formulation alongside the Sobolev framework. In contrast to prior Koopman-based analyses, which are primarily developed within Sobolev function classes, our approach examines how the choice of underlying function space influences the resulting complexity estimates. This provides a unified operator-theoretic perspective for studying both Sobolev- and Brownian/Cameron–Martin-based formulations.

## 3 Preliminaries

In this section, we assemble the foundational tools employed throughout the paper. Section 3.1 specifies the notation and measure-theoretic conventions; Section 3.2 reviews the notation of vector-valued Rademacher complexity; and Section 3.3 introduces the Koopman representation of the network.

### 3.1 Notations

In the following, we introduce a few notations that will be used throughout the paper. We denote the set of natural numbers by $\mathbb{N} := \{0, 1, 2, \ldots\}$, and the set of positive integers by $\mathbb{N}^* := \{1, 2, \ldots\}$. We denote $[-R, R]^d$ the $d$-dimensional hypercube for $R > 0$. The set of non-negative real numbers is written as $\mathbb{R}^{\geq 0}$. For a positive integer $n$, we define the set $[n]$ as $\{1, 2, \ldots, n\}$. The Cartesian product of a family of sets $(A_i)_{i \in I}$ is denoted by $\prod_{i \in I} A_i$. In particular, if $A_1 = \cdots = A_n = A$, we write $A^n$ for the $n$-fold Cartesian product of $A$ with itself. Let $\mathcal{U}$ be a topological space with a Borel sigma-field. We denote the space of probability measures on $\mathcal{U}$ as $\mathcal{P}(\mathcal{U})$. For a linear operator $\mathbf{W}$ on a Hilbert space, its range and kernel are denoted by $\mathrm{ran}(\mathbf{W})$ and $\ker(\mathbf{W})$, respectively. Its operator norm is denoted by $\|\mathbf{W}\|$. For an injective matrix $\mathbf{W} \in \mathbb{R}^{d \times d'}$ with $d \geq d'$, we define $|\det(\mathbf{W})| := \left(\det\left(\mathbf{W}^\top \mathbf{W}\right)\right)^{1/2}$. For a function $p \in L^\infty\left(\mathbb{R}^d\right)$, its $L^\infty$-norm is denoted by $\|p\|_\infty$. For a function $h$ on $\mathbb{R}^d$ and a subspace $\mathcal{S}$ of $\mathbb{R}^d$, the restriction of $h$ on $\mathcal{S}$ is denoted by $h|_{\mathcal{S}}$. With $\mathbb{S}_+^m \subset \mathbb{R}^{m \times m}$ we denote the set of $m \times m$ symmetric and positive semi-definite (p.s.d.) matrices. A function $k : \mathcal{X} \times \mathcal{X} \to \mathbb{R}$ is called a kernel if it is symmetric and positive semidefinite, that is, for every $n \in \mathbb{N}^*$, every collection of points $(\mathbf{x}_1, \ldots, \mathbf{x}_n) \in \mathcal{X}^n$, and all coefficients $(\alpha_1, \ldots, \alpha_n) \in \mathbb{R}^n$, $\sum_{i,j=1}^n \alpha_i \, k\left(\mathbf{x}_i, \mathbf{x}_j\right) \alpha_j \geq 0$. Each such kernel uniquely determines a reproducing kernel Hilbert space (RKHS) $\mathcal{H}_k \subset \mathbb{R}^{\mathcal{X}}$, consisting of real-valued functions on $\mathcal{X}$. The space $\mathcal{H}_k$ is characterized by the facts that $k_\mathbf{x}(\cdot) := k(\cdot, x) \in \mathcal{H}_k$, $\forall \mathbf{x} \in \mathcal{X}$, and it satisfies the reproducing property $h(\mathbf{x}) = \langle h, k_\mathbf{x} \rangle_{\mathcal{H}_k}$, $\forall h \in \mathcal{H}_k, \mathbf{x} \in \mathcal{X}$.[1] Finally, the RKHS admits the representation $\mathcal{H}_k = \overline{\mathrm{Span}\{k_\mathbf{x} : x \in \mathcal{X}\}}$, where span($\cdot$) denotes the linear span of its argument and $\overline{(\cdot)}$ indicates closure.

Let $\mathcal{X}$ be a non-empty set. A bivariate function $K : \mathcal{X} \times \mathcal{X} \to \mathbb{R}^{m \times m}$, $m \in \mathbb{N}$, is called a matrix-valued kernel if $K(\mathbf{x}, \mathbf{x}') = K(\mathbf{x}', \mathbf{x})^\top$ for all $(\mathbf{x}, \mathbf{x}') \in \mathcal{X}^2$, and for all $n \in \mathbb{N}$ and any $(\mathbf{x}_i, \mathbf{y}_i)_{i=1}^n \in (\mathcal{X} \times \mathcal{Y})^n$ we have $\sum_{i,j=1}^n \mathbf{y}_i^\top K(\mathbf{x}_i, \mathbf{x}_j)\mathbf{y}_j \geq 0$. Fix $m \in \mathbb{N}$ and let $\mathbf{M} \in \mathbb{S}_+^m$ be a symmetric positive semidefinite matrix. Let $K$ be a matrix-valued kernel. There exists a unique Hilbert space $\mathcal{H}_K$ of functions $f : \mathcal{X} \to \mathbb{R}^m$, called the vector-valued RKHS (vvRKHS) induced by $K$ (Micchelli & Pontil, 2005), such that for all $\mathbf{x} \in \mathcal{X}$, $\mathbf{y} \in \mathbb{R}^m$, and $f \in \mathcal{H}_K$, the function $\mathbf{x}' \mapsto K(\mathbf{x}, \mathbf{x}')\mathbf{y}$ belongs to $\mathcal{H}_K$ and $\langle f, K(\cdot, \mathbf{x})\mathbf{y} \rangle_{\mathcal{H}_K} = f(\mathbf{x})^\top \mathbf{y}$. In the separable case $K(\mathbf{x}, \mathbf{y}) = k(\mathbf{x}, \mathbf{y})\mathbf{M}$ with $\mathbf{M} \in \mathbb{S}_+^m$, the associated vvRKHS $\mathcal{H}_K$ consists of functions

---

[1] Here $k(\cdot, \mathbf{x})$ denotes the function $\mathbf{x}' \mapsto k(\mathbf{x}', \mathbf{x})$.

$f = (f_1, \ldots, f_m) : \mathcal{X} \to \mathbb{R}^m$ with each $f_j \in \mathcal{H}_k$, and the inner product is given by

$$\langle f, g \rangle_{\mathcal{H}_K} = \sum_{i,j=1}^{m} \left( \mathbf{M}^{-1} \right)_{ij} \langle f_i, g_j \rangle_{\mathcal{H}_k}, \qquad (\mathbf{M} \succ 0).$$

Equivalently,

$$\|f\|_{\mathcal{H}_K}^2 = \sum_{i,j=1}^{m} \left( \mathbf{M}^{-1} \right)_{ij} \langle f_i, f_j \rangle_{\mathcal{H}_k}.$$

If $\mathbf{M}$ is positive semidefinite (but not invertible), the same formulas hold on the corresponding quotient space induced by $\ker(\mathbf{M})$. In particular, the norm $\|f\|_{\mathcal{H}_K}$ is equivalent to the unweighted product norm $\|f\|_{\mathcal{H}_K}^2 \asymp \sum_{j=1}^{m} \|f_j\|_{\mathcal{H}_k}^2$, where the equivalence constants depend only on the eigenvalues of $\mathbf{M}$.

Let $\mathcal{H}_{K_1}$ and $\mathcal{H}_{K_2}$ be vvRKHSs on $\mathbb{R}^{d_1}$ and $\mathbb{R}^{d_2}$, induced by the matrix-valued kernels $K_1$ and $K_2$, respectively. Given a measurable function $f : \mathbb{R}^{d_1} \to \mathbb{R}^{d_2}$, the associated Koopman operator acts by composition from $\mathcal{H}_{K_2}$ to $\mathcal{H}_{K_1}$. We define its domain as $\mathcal{D}_f = \{g \in \mathcal{H}_{K_2} : g \circ f \in \mathcal{H}_{K_1}\}$, namely all functions in $\mathcal{H}_{K_2}$ whose pullback through $f$ belongs to $\mathcal{H}_{K_1}$. The Koopman operator associated with $f$ is the map $\mathcal{K}_f : \mathcal{D}_f \to \mathcal{H}_{K_1}$, $\mathcal{K}_f g = g \circ f$. For $\mathbf{b} \in \mathbb{R}^d$, the translation operator $T_{\mathbf{b}} : \mathbb{R}^d \to \mathbb{R}^d$ is $T_{\mathbf{b}}(x) := x + \mathbf{b}$, and the associated Koopman operator is $\mathcal{K}_{\mathbf{b}} g := g \circ T_{\mathbf{b}}$.

A multi-index is a vector $\alpha = (\alpha_1, \ldots, \alpha_d) \in \mathbb{N}^d$, with length $|\alpha| = \sum_{i=1}^{d} \alpha_i$. The symbol $D^\alpha f$ denotes the weak derivative of order $\alpha$. For $s \geq 0$, $H^s \left( \mathbb{R}^d \right)$ denotes the standard Sobolev space, identified with $W^{s,2} \left( \mathbb{R}^d \right)$. For a linear subspace $\mathcal{S} \subset \mathbb{R}^d$, we define $H^s \left( \mathcal{S}, \mathbb{R}^m \right)$ as the Sobolev space obtained by identifying $\mathcal{S}$ with $\mathbb{R}^{\dim \mathcal{S}}$ via an orthonormal basis. The notation $\widehat{f}$ denotes the Fourier transform of a function $f$, and $\text{supp}(f)$ its support. For $m \in \mathbb{N}$ and $s \geq 0$, $H^s \left( \mathbb{R}^d, \mathbb{R}^m \right)$ denotes the standard vector-valued Sobolev space, equipped with its usual Hilbert norm. The associated matrix-valued Sobolev kernel is written as $K_s \left( x, x' \right) = \boldsymbol{\Phi}_s \left( x - x' \right)$, where $\boldsymbol{\Phi}_s : \mathbb{R}^d \to \mathbb{R}^{m \times m}$ is a translation-invariant matrix-valued function whose Fourier transform satisfies the usual Sobolev decay condition. When $\boldsymbol{\Phi}_s$ is radial, we write $\boldsymbol{\Phi}_s \left( x \right) = \phi_s \left( \|x\|_2 \right)$ for a scalar profile function $\phi_s : [0, \infty) \to \mathbb{R}$.

Recall that the Brownian kernel on $\mathbb{R}$ is $k^{(\text{B})} \left( x, x' \right) = \frac{|x| + |x'| - |x - x'|}{2}$, $\forall x, x' \in \mathbb{R}$, which is non-negatively 1-homogeneous in the sense that $k^{(\text{B})} \left( ax, ax' \right) = a \, k^{(\text{B})} \left( x, x' \right)$ for all $a \in \mathbb{R}^{\geq 0}$.

Let $\mathcal{D} \subset \mathbb{R}^d$ be a domain, $\Omega$ a topological space equipped with a Borel probability measure $\mu \in \mathcal{P} \left( \Omega \right)$, and for each $\boldsymbol{\omega} \in \Omega$ let $k^{(\boldsymbol{\omega})} : \mathcal{D} \times \mathcal{D} \to \mathbb{R}$ be a scalar kernel with RKHS $\mathcal{H}_{k^{(\boldsymbol{\omega})}}$, satisfying $\int_\Omega k^{(\boldsymbol{\omega})} \left( \mathbf{x}, \mathbf{x} \right) \mathrm{d}\mu \left( \boldsymbol{\omega} \right) < \infty$, $\forall \mathbf{x} \in \mathcal{D}$. Define

$$\mathcal{H}_\oplus = \left\{ (f_{\boldsymbol{\omega}})_{\boldsymbol{\omega} \in \Omega} \in \prod_{\boldsymbol{\omega} \in \Omega} \mathcal{H}_{k^{(\boldsymbol{\omega})}} : \int_\Omega \|f_{\boldsymbol{\omega}}\|_{\mathcal{H}_{k^{(\boldsymbol{\omega})}}}^2 \, \mathrm{d}\mu \left( \boldsymbol{\omega} \right) < \infty \right\}, \tag{1}$$

with inner product $\langle f, g \rangle_{\mathcal{H}_\oplus} = \int_\Omega \langle f_{\boldsymbol{\omega}}, g_{\boldsymbol{\omega}} \rangle_{\mathcal{H}_{k^{(\boldsymbol{\omega})}}} \mathrm{d}\mu \left( \boldsymbol{\omega} \right)$. Then, by Hotz & Telschow (2012, Theorem 3.1), the space

$$\mathcal{H}_k = \left\{ f : \mathcal{D} \to \mathbb{R} : f \left( \mathbf{x} \right) = \int_\Omega f_{\boldsymbol{\omega}} \left( \mathbf{x} \right) \mathrm{d}\mu \left( \boldsymbol{\omega} \right) \ \forall \mathbf{x} \in \mathcal{D}, \ (f_{\boldsymbol{\omega}}) \in \mathcal{H}_\oplus \right\} \tag{2}$$

is an RKHS (the integral RKHS) with kernel $k \left( \mathbf{x}, \mathbf{y} \right) = \int_\Omega k^{(\boldsymbol{\omega})} \left( \mathbf{x}, \mathbf{y} \right) \mathrm{d}\mu \left( \boldsymbol{\omega} \right)$, $\mathbf{x}, \mathbf{y} \in \mathcal{D}$, and norm

$$\|f\|_{\mathcal{H}_k}^2 = \inf_{\substack{g \in \mathcal{H}_\oplus : \\ f(\mathbf{x}) = \int_\Omega g_{\boldsymbol{\omega}}(\mathbf{x}) \mathrm{d}\mu(\boldsymbol{\omega}), \ \forall \mathbf{x}}} \int_\Omega \|g_{\boldsymbol{\omega}}\|_{\mathcal{H}_{k^{(\boldsymbol{\omega})}}}^2 \, \mathrm{d}\mu \left( \boldsymbol{\omega} \right). \tag{3}$$

Let $\mathcal{X} \subset \mathbb{R}^d$ be a domain, $\mathbf{M} \in \mathbb{S}_+^m$, and let $K(\mathbf{x}, \mathbf{x}') := k^{(\text{B})}(\mathbf{x}, \mathbf{x}')\mathbf{M}$ be a separable Brownian kernel on $\mathcal{X} \times \mathcal{X}$. The induced vector-valued Brownian RKHS is denoted by $H^{(\text{B})} \left( \mathcal{X}, \mathbb{R}^m \right)$.

For an activation function $f : \mathbb{R} \to \mathbb{R}$, the sup-norm is $\|f\|_\infty := \sup_{x \in \mathbb{R}} |f(x)|$. For a function $\sigma : \mathbb{R}^d \to \mathbb{R}^d$ with component functions $(\sigma_i)_{i=1}^d$, the uniform derivative bound is $\|\sigma'\|_\infty := \max_{1 \leq i \leq d} \sup_{x \in \mathbb{R}^d} |\sigma_i'(x)|$, and analogously for $\left\| \left( \sigma^{-1} \right)' \right\|_\infty$.

## 3.2 Vector-valued Rademacher complexity

We consider a general multiple-output regression framework. Let us briefly recall the fundamental setting of supervised learning. We are given a training sample $\mathcal{D}_{XY,n} = \{(\mathbf{x}_i, \mathbf{y}_i)\}_{i=1}^n \sim \nu_{XY}^n$, where (i) $\nu_{XY} \in \mathcal{P}(\mathcal{X} \times \mathcal{Y})$ denotes the joint distribution governing the relationship between the input $X$ and the output $Y$; (ii) $\mathcal{X} \subset \mathbb{R}^d$ and $\mathcal{Y} \subset \mathbb{R}^m$ denote the input and output spaces, respectively; and (iii) $\nu_X$ represents the marginal distribution of $X$.[2] The objective is to learn a function $f$ from the data $\mathcal{D}_{XY,n}$ such that $f(x)$ provides an accurate prediction of the corresponding output $\mathbf{y}$ for unseen inputs $\mathbf{x}$. Based on the definition of Rademacher complexity, the vector-valued Rademacher complexity is defined as follows:

**Definition 1** (empirical vector-valued Rademacher complexity). *Let $\mathcal{F}$ be a class of functions $f : \mathcal{X} \to \mathbb{R}^m$ over an input space $\mathcal{X}$, and let*

$$\boldsymbol{\sigma}_i = (\sigma_{i1}, \ldots, \sigma_{im}) \sim \mathrm{Rad}^m, \quad i \in [n],$$

*be independent Rademacher vectors, i.e. $\sigma_{ij}$ are i.i.d. random variables uniformly distributed on $\{-1, +1\}$. Then, for a fixed dataset $\mathcal{D}_n = \{\mathbf{x}_i\}_{i=1}^n \subset \mathcal{X}$, the empirical vector-valued Rademacher complexity of $\mathcal{F}$ is defined as*

$$\widehat{\mathfrak{R}}_n^m(\mathcal{F}) := \mathbb{E}_{\mathrm{Rad}^{mn}} \left[ \sup_{f \in \mathcal{F}} \frac{1}{n} \left| \sum_{i=1}^n \langle \boldsymbol{\sigma}_i, f(\mathbf{x}_i) \rangle \right| \right], \tag{4}$$

*where $\langle \cdot, \cdot \rangle$ denotes the standard Euclidean inner product in $\mathbb{R}^m$. Equivalently, one may write $\boldsymbol{\sigma} = (\boldsymbol{\sigma}_1, \ldots, \boldsymbol{\sigma}_n) \sim \mathrm{Rad}^{nm}$.*

## 3.3 Koopman representation of deep networks

We study the generalization behaviour of an $L$-layer neural network

$$f = g \circ \mathbf{b}_L \circ \mathbf{W}_L \circ \sigma_{L-1} \circ \cdots \circ \sigma_1 \circ \mathbf{b}_1 \circ \mathbf{W}_1, \tag{5}$$

mapping $\mathbb{R}^{d_0}$ to $\mathbb{R}^m$. Here, $\mathbf{W}_l \in \mathbb{R}^{d_l \times d_{l-1}}$ are injective linear maps,

$$\mathbf{b}_l(x) = x + a_l, \qquad a_l \in \mathbb{R}^{d_l},$$

are bias shifts, $\sigma_l : \mathbb{R}^{d_l} \to \mathbb{R}^{d_l}$ are nonlinear activations, and $g : \mathbb{R}^{d_L} \to \mathbb{R}^m$ is the terminal representation map. Using the Koopman operator viewpoint, the network can be written as the operator product

$$f = \mathcal{K}_{\mathbf{W}_1} \mathcal{K}_{\mathbf{b}_1} \mathcal{K}_{\sigma_1} \cdots \mathcal{K}_{\mathbf{W}_{L-1}} \mathcal{K}_{\mathbf{b}_{L-1}} \mathcal{K}_{\sigma_{L-1}} \mathcal{K}_{\mathbf{W}_L} \mathcal{K}_{\mathbf{b}_L} \, g. \tag{6}$$

The Koopman representation separates the contribution of each layer at the level of function spaces: $\mathcal{K}_{\mathbf{W}_l}$ encodes the action of the linear map $\mathbf{W}_l$, $\mathcal{K}_{\mathbf{b}_l}$ accounts for translations, and $\mathcal{K}_{\sigma_l}$ captures the effect of the nonlinear activations. Rather than controlling generalization directly through the layer weights, we instead bound the corresponding Koopman operators acting on suitable vector-valued RKHSs. This operator-theoretic factorization exposes spectral-geometric quantities such as singular values and determinant factors in the resulting Rademacher complexity bounds. Each layer $\mathbb{R}^{d_l}$ is equipped with a vector-valued Sobolev RKHS $H^{s_l}(\mathbb{R}^{d_l}, \mathbb{R}^m)$, generated by the separable matrix-valued kernel

$$K_{s_l}(\mathbf{x}, \mathbf{x}') = k_{s_l}(\mathbf{x}, \mathbf{x}') \mathbf{M}, \qquad \text{for all } \mathbf{x}, \mathbf{x}' \in \mathcal{X}$$

where $k_{s_l}$ is a scalar Sobolev kernel of order $s_l > d_l/2$, and $\mathbf{M} \in \mathbb{S}_+^m$. Under these assumptions, the Koopman operators satisfy

$$\mathcal{K}_{\mathbf{W}_l} : H^{s_l}(\mathbb{R}^{d_l}, \mathbb{R}^m) \to H^{s_{l-1}}(\mathbb{R}^{d_{l-1}}, \mathbb{R}^m),$$
$$\mathcal{K}_{\mathbf{b}_l}, \, \mathcal{K}_{\sigma_l} : H^{s_l}(\mathbb{R}^{d_l}, \mathbb{R}^m) \to H^{s_l}(\mathbb{R}^{d_l}, \mathbb{R}^m),$$

which ensures that the Koopman factorization (6) is well-defined.

We impose the following assumption throughout this section.

---

[2] The notation $\nu_X$ will be used later when introducing Rademacher complexities.

**Assumption 1.** *The final nonlinear transformation $g \in H^{s_L}\left(\mathbb{R}^{d_L}, \mathbb{R}^m\right)$, and the Koopman operators $\mathcal{K}_{\sigma_l}$ are bounded for $l = 1, \ldots, L - 1$.*

To ensure that the terminal representation map satisfies Assumption 1, we may choose

$$g\left(\mathbf{x}\right) = e^{-\|\mathbf{x}\|^2}\mathbf{Mc}^\top, \qquad \mathbf{x} \in \mathbb{R}^{d_L},$$

for coefficients $\mathbf{c} \in \mathbb{R}^m$, matrix $\mathbf{M} \in \mathbb{S}_+^m$, and Sobolev order $s_L > d_L/2$. With this choice, $g \in H^{s_L}\left(\mathbb{R}^{d_L}, \mathbb{R}^m\right)$.

**Remark 1.** *Let $g$ be a smooth function which does not decay at infinity, for example the sigmoid activation. Although $H^{s_L}\left(\mathbb{R}^{d_L}, \mathbb{R}^m\right)$ does not contain such functions globally, one may construct $\widetilde{g} \in H^{s_L}\left(\mathbb{R}^{d_L}, \mathbb{R}^m\right)$ such that $\widetilde{g}\left(\mathbf{x}\right) = g\left(\mathbf{x}\right)$ on a sufficiently large compact region and replace $g$ by $\widetilde{g}$ in practical settings.*

**Assumption 2.** *There exists $\kappa > 0$ such that*

$$k_{s_0}\left(\mathbf{x}, \mathbf{x}\right) \leq \kappa, \qquad \forall \mathbf{x} \in \mathcal{X}.$$

Lemma B1 shows that under suitable smoothness and bi-Lipschitz assumptions on $\sigma_l$, the Koopman operator $\mathcal{K}_{\sigma_l}$ is bounded on $H^{s_l}\left(\mathbb{R}^{d_l}, \mathbb{R}^m\right)$, for $l \in [L - 1]$.

As activation functions, one may use smooth variants of Leaky ReLU, such as those introduced in Biswas et al. (2022).

We now derive Rademacher complexity bounds for the corresponding function class.

## 4 Koopman Formulation in Vector-Valued Sobolev RKHSs

We first derive Koopman-based Rademacher complexity bounds in vector-valued Sobolev RKHSs. The resulting estimates decompose into three components: a kernel-dependent factor associated with the input space, a product of Koopman operator norms corresponding to the nonlinear activations, and a product of spectral-geometric terms associated with the linear layers. The spectral-geometric contribution involves both operator norms and determinant factors. The determinant terms quantify the volume distortion induced by the linear maps, while the Sobolev-symbol ratios describe how regularity is transported across layers under the associated Koopman operators.

### 4.1 Koopman-based Sobolev bounds for invertible deep networks

In this subsection, we consider the case $d_l = d$, $l = 0, \ldots, L$, for some fixed $d \in \mathbb{N}$. For constants $C, D > 0$, define

$$\mathsf{W}(C, D) := \left\{\mathbf{W} \in \mathbb{R}^{d \times d} \mid \|\mathbf{W}\| \leq C, \ |\det\left(\mathbf{W}\right)| \geq D\right\},$$

and consider the invertible hypothesis class $\mathcal{F}_{\text{inv}} := \{f \in \mathcal{F} \mid \mathbf{W}_l \in \mathsf{W}(C, D)\}$. The following theorem gives a Koopman-based Rademacher complexity bound for $\mathcal{F}_{\text{inv}}$.

**Theorem 1.** *Assume that $\mathbf{M} \in \mathbb{S}_+^m$ and that the terminal map satisfies*

$$\|g\|_{H^{s_L}\left(\mathbb{R}^d, \mathbb{R}^m\right)} \leq B_g.$$

*Then the empirical Rademacher complexity of $\mathcal{F}_{\text{inv}}$ satisfies*

$$\widehat{\mathfrak{R}}_n^m\left(\mathcal{F}_{\text{inv}}\right) \leq B_g \sqrt{\frac{\kappa \operatorname{Tr}\left(\mathbf{M}\right)}{n}}$$

$$\cdot \sup_{\mathbf{W}_l \in \mathsf{W}(C, D)} \prod_{l=1}^{L} \sup_{\boldsymbol{\omega} \in \mathbb{R}^d} \left|\frac{\left(1 + \|\mathbf{W}_l^\top \boldsymbol{\omega}\|_2^2\right)^{s_{l-1}}}{\left(1 + \|\boldsymbol{\omega}\|_2^2\right)^{s_l}}\right|^{1/2} \frac{1}{|\det\left(\mathbf{W}_l\right)|^{1/2}} \prod_{l=1}^{L-1} \|\mathcal{K}_{\sigma_l}\|.$$

*Here $\kappa = \sup_{\mathbf{x} \in \mathbb{R}^d} k_{s_0}\left(\mathbf{x}, \mathbf{x}\right)$, and the translation Koopman operators satisfy $\|\mathcal{K}_{\mathbf{b}_l}\| = 1$ on the Sobolev spaces considered here.*

The bound in Theorem 1 follows from the Koopman factorization of the network and the corresponding layer-wise operator estimates. In particular, the determinant terms arise from the change-of-variables structure of the linear Koopman operators and quantify the volume distortion induced by the weight matrices. Applying Lemma 5 of Hashimoto et al. (2024) to Theorem 1 yields the following corollary.

**Corollary 1.** *Assume*

$$H^{s_l}\left(\mathbb{R}^d, \mathbb{R}^m\right) = H^s\left(\mathbb{R}^d, \mathbb{R}^m\right), \qquad l = 0, \ldots, L,$$

*for some $s > d/2$. Then*

$$\widehat{\mathfrak{R}}_n^m\left(\mathcal{F}_{\text{inv}}\right) \leq \left(\sqrt{\frac{\kappa \operatorname{Tr}(\mathbf{M})}{n}}\right) \|g\|_{H^s(\mathbb{R}^d, \mathbb{R}^m)} \left(\frac{\max\{1, C^s\}}{D^{1/2}}\right)^L \prod_{l=1}^{L-1} \|\mathcal{K}_{\sigma_l}\|.$$

### 4.2 Koopman-based Sobolev bounds for injective deep networks

We now extend the invertible-layer analysis to injective architectures with width expansion, where the linear maps $\mathbf{W}_l : \mathbb{R}^{d_{l-1}} \to \mathbb{R}^{d_l}$ are injective but not necessarily square. In this setting, the determinant of $\mathbf{W}_l$ is no longer defined. The relevant geometric quantity is instead $\det\left(\mathbf{W}_l^\top \mathbf{W}_l\right)$, which measures the volume distortion induced by $\mathbf{W}_l$ on its range. A second feature of the injective setting is the appearance of geometric factors $\mathrm{G}_l$, which quantify how much of the function mass is concentrated on the lower-dimensional subspace $\operatorname{ran}(\mathbf{W}_l) \subset \mathbb{R}^{d_l}$. These factors capture the interaction between Sobolev regularity and the geometry induced by the injective linear layers. For constants $C, D > 0$, define

$$\mathsf{W}_l(C, D) := \left\{ \mathbf{W} \in \mathbb{R}^{d_l \times d_{l-1}} \mid d_l \geq d_{l-1}, \ \|\mathbf{W}\| \leq C, \ \det\left(\mathbf{W}^\top \mathbf{W}\right)^{1/2} \geq D \right\}.$$

Since $\mathbf{W} : \mathbb{R}^{d_{l-1}} \to \mathbb{R}^{d_l}$ is injective, the matrix $\mathbf{W}^\top \mathbf{W}$ is a positive-definite $d_{l-1} \times d_{l-1}$ matrix. Consequently, $\det\left(\mathbf{W}^\top \mathbf{W}\right)$ is well-defined and measures the squared volume distortion induced by $\mathbf{W}$ on its image. Consider the injective hypothesis class $\mathcal{F}_{\text{inj}} := \{f \in \mathcal{F} \mid \mathbf{W}_l \in \mathsf{W}_l(C, D)\}$. For

$$f_l := g \circ \mathbf{b}_L \circ \mathbf{W}_L \circ \sigma_{L-1} \circ \mathbf{b}_{L-1} \circ \mathbf{W}_{L-1} \circ \cdots \circ \sigma_l \circ \mathbf{b}_l,$$

an injective linear map $\mathbf{W}_l : \mathbb{R}^{d_{l-1}} \to \mathbb{R}^{d_l}$, let $R_{\mathbf{W}_l} h = h|_{\operatorname{ran}(\mathbf{W}_l)}$ denote the restriction operator.

Throughout this subsection, we assume that the restriction $R_{\mathbf{W}_l}$ is well-defined and bounded as a map

$$R_{\mathbf{W}_l} : H^{s_l}\left(\mathbb{R}^{d_l}, \mathbb{R}^m\right) \longrightarrow H^{s_{l-1}}\left(\operatorname{ran}(\mathbf{W}_l), \mathbb{R}^m\right),$$

for every layer $l$. This assumption is satisfied whenever the Sobolev exponents are chosen so that the standard Sobolev trace/restriction theorem applies to the linear subspace $\operatorname{ran}(\mathbf{W}_l) \subset \mathbb{R}^{d_l}$. Define

$$\mathrm{G}_l := \frac{\|R_{\mathbf{W}_l} f_l\|_{H^{s_{l-1}}(\operatorname{ran}(\mathbf{W}_l), \mathbb{R}^m)}}{\|f_l\|_{H^{s_l}(\mathbb{R}^{d_l}, \mathbb{R}^m)}}.$$

The following theorem gives the corresponding injective Koopman-based Rademacher complexity bound.

**Theorem 2.** *Assume that $\mathbf{M} \in \mathbb{S}_+^m$ and that the terminal map satisfies*

$$\|g\|_{H^{s_L}\left(\mathbb{R}^{d_L}, \mathbb{R}^m\right)} \leq B_g.$$

*The Rademacher complexity $\widehat{\mathfrak{R}}_n^m\left(\mathcal{F}_{\text{inj}}\right)$ satisfies*

$$\widehat{\mathfrak{R}}_n^m\left(\mathcal{F}_{\text{inj}}\right) \leq B_g \left(\sqrt{\frac{\kappa \operatorname{Tr}(\mathbf{M})}{n}}\right) \|g\|_{H^{s_L}\left(\mathbb{R}^{d_L}, \mathbb{R}^m\right)}$$

$$\cdot \sup_{\mathbf{W}_l \in \mathsf{W}_l(C, D)} \prod_{l=1}^L \mathrm{G}_l \sup_{\boldsymbol{\omega} \in \operatorname{ran}(\mathbf{W}_l)} \left| \frac{1 + \|\mathbf{W}_l^\top \boldsymbol{\omega}\|_2^2}{1 + \|\boldsymbol{\omega}\|_2^2} \right|^{s_{l-1}/2} \frac{1}{\det\left(\mathbf{W}_l^\top \mathbf{W}_l\right)^{1/4}} \prod_{l=1}^{L-1} \|\mathcal{K}_{\sigma_l}\|.$$

**Remark 2** (Non-injective weight matrices). *The injectivity assumption in Theorem 2 guarantees bound-edness of the composition operator associated with $\mathbf{W}_l$. When $\mathbf{W}_l$ is rank-deficient, directions in $\ker(\mathbf{W}_l)$ collapse, and the corresponding Koopman operator is no longer bounded on Sobolev-type RKHSs. Follow-ing the operator-regularization viewpoint introduced in Hashimoto et al. (2024, Section 4.3.1), one can re-place the singular volume factor by the stabilized quantity $\det\left(I + \mathbf{W}_l^\top \mathbf{W}_l\right)$, which remains strictly posi-tive even for rank-deficient layers. This leads to the modified estimate $1/\det\left(I + \mathbf{W}_l^\top \mathbf{W}_l\right)^{1/4}$ in place of $1/\det\left(\mathbf{W}_l^\top \mathbf{W}_l\right)^{1/4}$. The resulting bound remains finite in the non-injective setting and preserves the same operator-theoretic interpretation in terms of regularized volume distortion.*

## 5  Koopman Formulation in Vector-Valued Brownian RKHSs

We now consider an alternative function-space setting based on the one-dimensional Brownian/Cameron–Martin RKHS. In contrast to the Sobolev framework of the previous section, where regularity is measured through Sobolev norms of order $s_l$, the Brownian setting is governed by a first-order Cameron–Martin structure. This leads to a different Koopman-operator analysis and a different form of complexity control.

More precisely, we work with the classical one-dimensional Brownian kernel and its associated vector-valued RKHS. By the Cameron–Martin characterization, the resulting norm is equivalent to a first-order derivative/Fourier-energy seminorm. Consequently, the effect of linear and nonlinear transformations can be analyzed directly through the corresponding Koopman operators acting on this first-order geometry.

The resulting complexity bound differs qualitatively from the Sobolev bounds of Section 4.1. In particular, the dependence on the linear layers is governed by the one-dimensional Brownian scaling law, yielding factors of the form $|W_l|^{1/2}$, while the activation operators are controlled through Cameron–Martin composition estimates. As a result, the Brownian/Cameron–Martin framework provides an alternative function-space perspective on Koopman-based generalization analysis and leads to a distinct spectral structure from the Sobolev setting.

The following theorem establishes the corresponding Brownian/Cameron–Martin Rademacher complexity bound for invertible architectures.

### 5.1  Koopman-based Brownian bounds for invertible networks

In this subsection, we specialize to the one-dimensional Brownian/Cameron–Martin setting. Let $R > 0$ and define $\mathcal{X} = [-R, R]$. Let

$$K\left(x, x'\right) = k^{(\mathrm{B})}\left(x, x'\right)\mathbf{M}, \qquad x, x' \in \mathcal{X},$$

where

$$k^{(\mathrm{B})}\left(x, x'\right) = \frac{|x| + |x'| - |x - x'|}{2}$$

is the one-dimensional Brownian kernel and $\mathbf{M} \in \mathbb{S}_+^m$. We denote by $H^{(\mathrm{B})}\left(\mathcal{X}, \mathbb{R}^m\right)$ the associated vector-valued Brownian RKHS. By Lemma B3, this space is norm-equivalent to the vector-valued Cameron–Martin space on $[-R, R]$. Throughout this subsection, we assume that all functions admit compactly supported Sobolev extensions to $\mathbb{R}$. For constants (C>D>0), define

$$\mathsf{W}(C, D) := \{W \in \mathbb{R} : D \le |W| \le C\}.$$

We consider the Brownian hypothesis class

$$\mathcal{F}_{\mathrm{inv}}^{(\mathrm{B})} := \left\{f = \mathcal{K}_{W_1}\mathcal{K}_{\sigma_1}\cdots\mathcal{K}_{W_L}g \ \middle| \ g \in H^{(\mathrm{B})}\left(\mathcal{X}, \mathbb{R}^m\right), \ \|g\|_{H^{(\mathrm{B})}(\mathcal{X}, \mathbb{R}^m)} \le B_g, \ W_l \in \mathsf{W}(C, D)\right\},$$

where $\sigma_l$ satisfy the assumptions of Lemma B4.

The following theorem gives the corresponding Brownian/Cameron–Martin Koopman complexity bound.

**Theorem 3.** *Assume that $\mathbf{M} \in \mathbb{S}_+^m$ and that $\|g\|_{H^{(\mathrm{B})}(\mathcal{X}, \mathbb{R}^m)} \leq B_g$. Let $C_{\mathrm{lin}} := \frac{c_2}{c_1}$, where $c_1, c_2$ are the norm-equivalence constants from Lemma B3, and let $C_{\mathrm{B}}$ be the activation constant from Lemma B4. Then the empirical Rademacher complexity satisfies*

$$\widehat{\mathfrak{R}}_n^m \left( \mathcal{F}_{\mathrm{inv}}^{(\mathrm{B})} \right) \leq B_g \sqrt{\frac{\kappa \operatorname{Tr}(\mathbf{M})}{n}} \, C_{\mathrm{lin}}^L C_{\mathrm{B}}^{L-1} \sup_{W_l \in \mathsf{W}(C,D)} \prod_{l=1}^{L} |W_l|^{1/2} \prod_{l=1}^{L-1} \|\sigma_l'\|_\infty \|(\sigma_l^{-1})'\|_\infty^{1/2}.$$

*Here $\kappa = \sup_{x \in \mathcal{X}} k^{(\mathrm{B})}(x, x) \leq R$.*

**Remark 3.** *The Brownian/Cameron–Martin formulation considered in this section should be viewed as an alternative function-space regime rather than a refinement of the Sobolev analysis of Hashimoto et al. (2024). The two approaches are based on different underlying RKHSs and therefore lead to different Koopman operator estimates.*

*In the present work, we specialize to the one-dimensional Brownian/Cameron–Martin RKHS generated by the Brownian kernel The resulting complexity bound is therefore not directly comparable to the multidimensional Sobolev bounds of Hashimoto et al. (2024), which are formulated in Sobolev RKHSs of order $s > d/2$.*

*Nevertheless, a notable difference appears in the activation-operator estimates. In the one-dimensional Cameron–Martin setting, we obtain*

$$\|\mathcal{K}_\sigma\| \leq C_{\mathrm{B}} \|\sigma'\|_\infty \|(\sigma^{-1})'\|_\infty^{1/2},$$

*whereas the corresponding Sobolev $(s = 1)$ estimate of Hashimoto et al. (2024) takes the form*

$$\|\mathcal{K}_\sigma\| \lesssim \|(\sigma^{-1})'\|_\infty \max\{1, \|\sigma'\|_\infty\}.$$

*Thus, while the two analyses operate on different hypothesis spaces and should not be interpreted as directly comparable complexity bounds, the Brownian/Cameron–Martin estimate exhibits a milder dependence on the inverse-derivative factor. In particular, if $\varepsilon := \inf_x |\sigma'(x)| > 0$, then*

$$\|(\sigma^{-1})'\|_\infty = \frac{1}{\inf_x \sigma'(x)} = \frac{1}{\varepsilon}.$$

*Consequently, the Sobolev estimate scales proportionally to $1/\varepsilon$, whereas the Brownian/Cameron–Martin estimate scales proportionally to $1/\sqrt{\varepsilon}$. This distinction becomes most visible for invertible activations that contain nearly flat regions, where the inverse derivative can become large.*

## 6 Shared Operator Learning and Transfer

In the previous sections, we studied operator-theoretic generalization bounds for fixed Koopman composition chains acting on vector-valued RKHSs. We now consider a related extension in which a shared operator is learned jointly across multiple tasks.

The purpose of this section is not to introduce a new empirical transfer-learning framework, but rather to show that the operator-theoretic viewpoint naturally admits a shared operator-learning formulation with an exact finite-dimensional representer structure. In contrast to multitask formulations based solely on output kernels or finite-dimensional latent representations, the transferable object considered here is an operator acting between Hilbert function spaces.

The results below are inspired by operator-learning representer methodologies for Koopman operators in scalar RKHS settings (Khosravi, 2023). However, the present formulation differs in that it operates in vector-valued RKHSs and studies shared multitask transfer operators motivated by the Koopman product framework developed earlier in this paper.

## 6.1 Shared operator-learning formulation

Let $\mathcal{H}_{\text{out}}$ be a Hilbert space containing the final output-side maps $g$ appearing in the Koopman factorization. Thus, $\mathcal{H}_{\text{out}}$ does not refer to the penultimate hidden-layer activations, but to the function space in which the terminal map $g : \mathbb{R}^{d_L} \to \mathbb{R}^m$ is assumed to lie. In the present framework, one may take for example $\mathcal{H}_{\text{out}} = H^{s_L}\left(\mathbb{R}^{d_L}, \mathbb{R}^m\right)$ or $\mathcal{H}_{\text{out}} = H^{(\text{B})}\left(\mathcal{X}, \mathbb{R}^m\right)$. In the present framework, one may take for example $\mathcal{H}_{\text{out}} = H^{s_L}\left(\mathbb{R}^{d_L}, \mathbb{R}^m\right)$ or $\mathcal{H}_{\text{out}} = H^{(\text{B})}\left(\mathcal{X}, \mathbb{R}^m\right)$. Let $\mathcal{H}_K$ be a vector-valued RKHS induced by a matrix-valued kernel $K : \mathcal{X} \times \mathcal{X} \to \mathbb{R}^{m \times m}$. We consider a shared operator $T : \mathcal{H}_{\text{out}} \to \mathcal{H}_K$. Motivated by the Koopman factorization introduced earlier, one may formally view $T$ as being induced by products of the form $\mathcal{K}_{\mathbf{W}_1} \mathcal{K}_{\mathbf{b}_1} \mathcal{K}_{\sigma_1} \cdots \mathcal{K}_{\mathbf{W}_L} \mathcal{K}_{\mathbf{b}_L}$. Let $M \in \mathbb{N}$ denote the number of source tasks. For each $t \in [M]$, let $g_t \in \mathcal{H}_{\text{out}}$ denote a task-specific terminal representation, and let $(\mathbf{x}_{ti}, \mathbf{y}_{ti}) \in \mathcal{X} \times \mathbb{R}^m$, $i \in [n_t]$, be the corresponding training samples. We study the regularized operator-learning problem

$$\min_T \sum_{t=1}^M \sum_{i=1}^{n_t} \ell\left(\mathbf{y}_{ti}, (Tg_t)(\mathbf{x}_{ti})\right) + \lambda \|T\|_{\text{HS}}^2, \tag{7}$$

where $T : \mathcal{H}_{\text{out}} \to \mathcal{H}_K$, $\lambda > 0$, the loss $\ell$ is convex and continuous in its second argument, and $\|\cdot\|_{\text{HS}}$ denotes the Hilbert–Schmidt norm.

## 6.2 Representer theorem

The next theorem shows that every minimum-norm minimizer of (7) admits a finite-rank representation determined entirely by the training samples and the task representations.

**Theorem 4.** *Assume that* (7) *admits a minimizer. Then every minimum-norm minimizer* $\widehat{T} : \mathcal{H}_{\text{out}} \to \mathcal{H}_K$ *admits the representation*

$$\widehat{T} = \sum_{t=1}^M \sum_{i=1}^{n_t} \sum_{a=1}^m c_{tia} \left(K\left(\cdot, \mathbf{x}_{ti}\right) \mathbf{e}_a\right) \otimes g_t.$$

*where* $(\mathbf{e}_a)_{a=1}^m$ *is the standard basis of* $\mathbb{R}^m$. *Equivalently,*

$$\left(\widehat{T}g\right)(\mathbf{x}) = \sum_{t=1}^M \sum_{i=1}^{n_t} \sum_{a=1}^m c_{tia} \langle g_t, g\rangle_{\mathcal{H}_{\text{out}}} K\left(\mathbf{x}, \mathbf{x}_{ti}\right) \mathbf{e}_a.$$

## 6.3 Finite-dimensional reduction

The representer theorem implies that the infinite-dimensional optimization problem (7) reduces exactly to a finite-dimensional optimization problem over the coefficients $(c_{tia})$.

**Theorem 5.** *Let* $\widehat{T}$ *be a minimum-norm minimizer of* (7). *Then the corresponding coefficients solve the finite-dimensional optimization problem*

$$\min_c \sum_{t=1}^M \sum_{i=1}^{n_t} \left\| \mathbf{y}_{ti} - \sum_{t'=1}^M \sum_{j=1}^{n_{t'}} \sum_{a=1}^m c_{t'ja} \langle g_{t'}, g_t\rangle_{\mathcal{H}_{\text{out}}} K\left(\mathbf{x}_{ti}, \mathbf{x}_{t'j}\right) \mathbf{e}_a \right\|_2^2$$
$$+ \lambda \sum_{\substack{t,i,a \\ t',j,b}} c_{tia} c_{t'jb} \langle g_t, g_{t'}\rangle_{\mathcal{H}_{\text{out}}} \mathbf{e}_a^\top K\left(\mathbf{x}_{ti}, \mathbf{x}_{t'j}\right) \mathbf{e}_b.$$

## 6.4 Transfer bound

We finally record the generalization consequence of the learned shared operator. Let $K(\mathbf{x}, \mathbf{x}') = k(\mathbf{x}, \mathbf{x}')\mathbf{M}$, $\mathbf{M} \in \mathbb{S}_+^m$, with $k(\mathbf{x}, \mathbf{x}) \leq \kappa$. Conditionally on the learned operator $\widehat{T}$, define $\mathcal{F}_{\widehat{T}}(B) \coloneqq \left\{\widehat{T}g : \|g\|_{\mathcal{H}_{\text{out}}} \leq B\right\}$. Let

$$R_0(f) \coloneqq \mathbb{E}\left[\ell(Y, f(X))\right]$$

denote the target population risk, and let $f_0^\star \in \arg\min_f R_0(f)$. Let $\widehat{f}_0$ be an empirical risk minimizer over $\mathcal{F}_{\widehat{T}}(B)$ on a target sample of size $n_0$, and define

$$A_{\widehat{T}}(B) \coloneqq \inf_{\|g\|_{\mathcal{H}_{\mathrm{out}}} \leq B} R_0\left(\widehat{T}g\right) - R_0\left(f_0^\star\right).$$

**Proposition 1.** *Assume that the loss is $L_\ell$-Lipschitz and bounded by $C_\ell$. Then, conditionally on $\widehat{T}$, with probability at least $1 - \delta$,*

$$R_0\left(\widehat{f}_0\right) - R_0\left(f_0^\star\right) \leq A_{\widehat{T}}(B) + 4 L_\ell B \left\|\widehat{T}\right\| \sqrt{\frac{\kappa \operatorname{Tr}(\mathbf{M})}{n_0}} + C_\ell \sqrt{\frac{\log(1/\delta)}{2 n_0}}.$$

*In particular, since $\left\|\widehat{T}\right\| \leq \left\|\widehat{T}\right\|_{\mathrm{HS}}$, the same bound holds with $\left\|\widehat{T}\right\|_{\mathrm{HS}}$ in place of $\left\|\widehat{T}\right\|$.*

**Remark 4.**

- *Theorem 1 does not imply that shared operator learning universally improves transfer performance. It shows only that whenever a useful shared operator admits controlled operator norm, the induced transfer class inherits a corresponding Rademacher complexity bound.*

- *The present viewpoint differs from multitask formulations based solely on output kernels or finite-dimensional latent representations. Here, the transferable object is an operator acting between Hilbert function spaces.*

- *When the learned operator $\widehat{T}$ is induced by Koopman composition chains satisfying the bounds of Theorems 1 to 3, the transfer complexity term $\left\|\widehat{T}\right\|$ inherits the corresponding spectral-geometric structure involving Koopman operator norms, determinant factors, and Brownian/Sobolev function-space geometry.*

## 7 Experimental Details

In this section, we present an empirical comparison between the Sobolev-based and Brownian-inspired complexity factors introduced in our analysis. We first conduct a numerical study of the behavior of these quantities in a synthetic regression setting. We then investigate their use as regularization mechanisms in a multi-class classification task, analyzing their impact on predictive performance and generalization.

### 7.1 Numerical comparison of the complexity factors

To numerically evaluate the bound, we conducted a synthetic data experiment inspired by Hashimoto et al. (2024). The synthetic data consider a regression problem in $\mathbb{R}^3$, where the target function $t$ is defined as $t(x) \coloneqq \exp\left(-\|2x - \mathbf{1}\|^2\right)$, for all $x \in \mathbb{R}^3$. We employ a fully connected neural network of the form $f(x) \coloneqq g\left(W_2\, \sigma\left(W_1 x + b_1\right) + b_2\right)$, with parameter matrices and vectors $W_1 \in \mathbb{R}^{3\times3}$, $b_1 \in \mathbb{R}^3$, $W_2 \in \mathbb{R}^{6\times3}$, and $b_2 \in \mathbb{R}^6$. The matrices $W_1$ and $W_2$ are initialized using orthogonal initialization via Saxe et al. (2014), and the bias vectors $b_1$ and $b_2$ are initialized by sampling from a uniform distribution. As activation function we use the smooth Leaky ReLU from Biswas et al. (2022), and for the output mapping we take $g(x) \coloneqq \exp\left(-\|x\|^2\right)$. The network is trained for 1600 epochs using gradient-based optimization with learning rate $3 \times 10^{-3}$, and an $L_2$ penalty of $10^{-4}$ is employed. The Sobolev Bound is defined as $\mathrm{SB} \coloneqq \prod_{l=1}^{L} \|W_l\|^{s_l} / \left(\det\left(I + W_l^\top W_l\right)\right)^{1/4}$, for $s_l \coloneqq \frac{d_l + 0.1}{2}$, while the Brownian Bound is given by $\mathrm{BB} \coloneqq \prod_{l=1}^{L} \|W_l\| / \left(\det\left(I + W_l^\top W_l\right)\right)^{1/4}$. Figure 1(a) displays the numerical values of these two bounds, evaluated throughout training across five independent weight initializations. As training progresses, the Brownian-inspired complexity factor exhibits a different empirical behavior from the Sobolev-based factor, remaining numerically smaller and displaying less variability across the considered random initializations. These observations are consistent with the distinct function-space perspectives underlying the two analyses. Additional experimental details are reported in Appendix D.

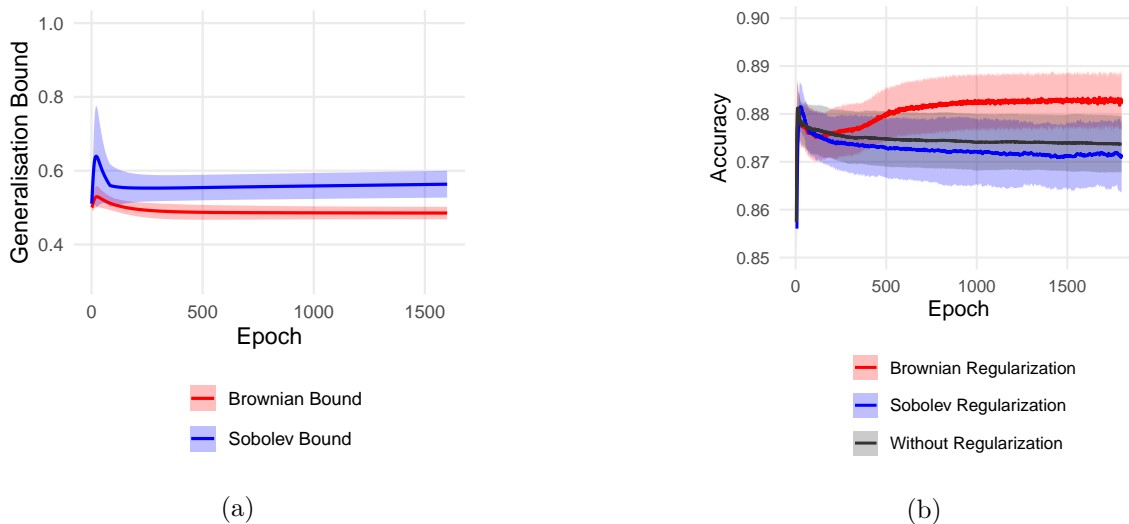

Figure 1: (a) Evolution of the Sobolev-based and Brownian-inspired complexity factors throughout training on the synthetic dataset. (b) Test performance on MNIST comparing models regularized via the Sobolev Bound, via the Brownian Bound, and a baseline without regularization.

## 8 Limitations

The theoretical results developed in this work rely on several assumptions that delimit their scope of applicability. First, the Koopman-based bounds require either invertible linear layers or injective linear maps satisfying the geometric conditions used in Sections 4.1 and 4.2. For rank-deficient architectures, additional regularization mechanisms are required to obtain finite operator bounds. Second, the activation analyses require sufficient regularity of the nonlinearities. In particular, the Sobolev results assume activations whose associated Koopman operators are bounded on the underlying Sobolev spaces, while the Brownian/Cameron–Martin analysis is currently developed in a one-dimensional setting and assumes invertible activations with controlled derivatives and inverse derivatives in order to establish the corresponding Koopman composition estimates. Third, several arguments rely on domain-control assumptions. These include compact domains, bounded input regions, or the existence of suitable extensions of the functions under consideration. Such assumptions are standard in RKHS-based analyses but may not hold globally for arbitrary deep-network architectures. Finally, the framework assumes that the terminal representation map belongs to a prescribed function space and satisfies an explicit norm constraint. The resulting complexity estimates therefore depend on the choice of this terminal map and do not provide a completely architecture-independent characterization.

## 9 Conclusion

We developed an operator-theoretic framework for deriving Rademacher complexity bounds for deep multi-output networks via Koopman operators acting on vector-valued RKHSs. Our results extend existing Koopman-based Sobolev analyses to vector-valued settings and to non-square architectures, and investigate a Brownian/Cameron–Martin function-space formulation that provides an alternative operator-theoretic perspective to the Sobolev setting. We additionally introduced a shared operator-learning formulation for multitask transfer in vector-valued RKHSs, deriving an exact representer theorem, a finite-dimensional reduction of the corresponding operator-learning problem, and transfer bounds for the induced operator class. More broadly, the results highlight how changing the underlying function space modifies the geometry and spectral behaviour of Koopman-based generalization estimates.

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

# A    Additional Notations

We collect here several additional operator-theoretic and functional-analytic notations used throughout the appendices and proofs.

Let $\mathcal{H}_1, \mathcal{H}_2$ be Hilbert spaces. We denote by $\mathcal{L}(\mathcal{H}_1, \mathcal{H}_2)$ the space of bounded linear operators from $\mathcal{H}_1$ to $\mathcal{H}_2$, equipped with the operator norm

$$\|T\| := \sup_{\|f\|_{\mathcal{H}_1} \leq 1} \|Tf\|_{\mathcal{H}_2}.$$

The space of Hilbert–Schmidt operators from $\mathcal{H}_1$ to $\mathcal{H}_2$ is denoted by $\mathcal{L}_2(\mathcal{H}_1, \mathcal{H}_2)$, equipped with the Hilbert–Schmidt norm

$$\|T\|_{\mathrm{HS}} := \left( \sum_{j=1}^{\infty} \|Te_j\|_{\mathcal{H}_2}^2 \right)^{1/2},$$

where $(e_j)_{j \in \mathbb{N}}$ is an arbitrary orthonormal basis of $\mathcal{H}_1$. For $u \in \mathcal{H}_2$ and $g \in \mathcal{H}_1$, the rank-one operator $u \otimes g \in \mathcal{L}_2(\mathcal{H}_1, \mathcal{H}_2)$ is defined by

$$(u \otimes g)h = u \langle g, h \rangle_{\mathcal{H}_1}, \qquad h \in \mathcal{H}_1.$$

For a subset $A$ of a vector space, we denote by $\mathrm{Span}(A)$ its linear span. For a matrix $A$, the trace is denoted by $\mathrm{Tr}(A)$, and the diagonal matrix generated by a vector $v$ is written as $\mathrm{diag}(v)$. For a closed subspace $V$ of a Hilbert space $\mathcal{H}$, the orthogonal projection onto $V$ is denoted by $P_V : \mathcal{H} \to V$. Finally, throughout the appendices, the notation $\widehat{f}$ denotes the Fourier transform of $f$.

# B    Proofs

All proofs of Theorems 1 to 5 and Proposition 1 appear in the main text.

## B.1    Proof of Theorem 1

We first estimate the Koopman operators associated with the translation and linear layers.

*Translation operators.* Let $d_l = d$ for $l = 0, \ldots, L$. Let $h \in H^{s_l}(\mathbb{R}^d, \mathbb{R}^m)$. Using the shift-invariance property of the Fourier transform,

$$\widehat{h \circ \mathbf{b}_l}(\boldsymbol{\omega}) = e^{-i\mathbf{a}_l^\top \boldsymbol{\omega}} \widehat{h}(\boldsymbol{\omega}),$$

where (a) follows from the Fourier transform of translations. Consequently,

$$
\begin{aligned}
\|\mathcal{K}_{\mathbf{b}_l} h\|_{H^{s_l}(\mathbb{R}^d, \mathbb{R}^m)}^2 &\overset{(a)}{=} \int_{\mathbb{R}^d} \left(1 + \|\boldsymbol{\omega}\|_2^2\right)^{s_l} \left\|\widehat{h \circ \mathbf{b}_l}(\boldsymbol{\omega})\right\|_2^2 \mathrm{d}\boldsymbol{\omega} \\
&\overset{(b)}{=} \int_{\mathbb{R}^d} \left(1 + \|\boldsymbol{\omega}\|_2^2\right)^{s_l} \left|e^{-i\mathbf{a}_l^\top \boldsymbol{\omega}}\right|^2 \left\|\widehat{h}(\boldsymbol{\omega})\right\|_2^2 \mathrm{d}\boldsymbol{\omega} \\
&\overset{(c)}{=} \|h\|_{H^{s_l}(\mathbb{R}^d, \mathbb{R}^m)}^2,
\end{aligned}
$$

where (a) follows from the definition of the Sobolev norm, (b) uses the previous identity, and (c) follows from $\left|e^{-i\mathbf{a}_l^\top \boldsymbol{\omega}}\right| = 1$. Hence, $\|\mathcal{K}_{\mathbf{b}_l}\| = 1$.

*Linear layers.* Let $h \in H^{s_l}(\mathbb{R}^d, \mathbb{R}^m)$. Using the scaling property of the Fourier transform,

$$
\widehat{h \circ \mathbf{W}_l}(\boldsymbol{\omega}) = \frac{1}{|\det(\mathbf{W}_l)|} \widehat{h}\left(\mathbf{W}_l^{-\top} \boldsymbol{\omega}\right),
$$

Therefore,

$$
\begin{aligned}
\|\mathcal{K}_{\mathbf{W}_l} h\|_{H^{s_{l-1}}(\mathbb{R}^d, \mathbb{R}^m)}^2 &\overset{(a)}{=} \int_{\mathbb{R}^d} \left(1 + \|\boldsymbol{\omega}\|_2^2\right)^{s_{l-1}} \left\|\widehat{h \circ \mathbf{W}_l}(\boldsymbol{\omega})\right\|_2^2 \mathrm{d}\boldsymbol{\omega} \\
&\overset{(b)}{=} \frac{1}{|\det(\mathbf{W}_l)|} \int_{\mathbb{R}^d} \left(1 + \|\boldsymbol{\omega}\|_2^2\right)^{s_{l-1}} \left\|\widehat{h}\left(\mathbf{W}_l^{-\top} \boldsymbol{\omega}\right)\right\|_2^2 \mathrm{d}\boldsymbol{\omega} \\
&\overset{(c)}{=} \frac{1}{|\det(\mathbf{W}_l)|} \int_{\mathbb{R}^d} \frac{\left(1 + \|\mathbf{W}_l^\top \boldsymbol{\xi}\|_2^2\right)^{s_{l-1}}}{\left(1 + \|\boldsymbol{\xi}\|_2^2\right)^{s_l}} \left(1 + \|\boldsymbol{\xi}\|_2^2\right)^{s_l} \left\|\widehat{h}(\boldsymbol{\xi})\right\|_2^2 \mathrm{d}\boldsymbol{\xi} \\
&\overset{(d)}{\leq} \sup_{\boldsymbol{\xi} \in \mathbb{R}^d} \left|\frac{\left(1 + \|\mathbf{W}_l^\top \boldsymbol{\xi}\|_2^2\right)^{s_{l-1}}}{\left(1 + \|\boldsymbol{\xi}\|_2^2\right)^{s_l}}\right| \frac{1}{|\det(\mathbf{W}_l)|} \|h\|_{H^{s_l}(\mathbb{R}^d, \mathbb{R}^m)}^2,
\end{aligned}
$$

where (a) follows from the definition of the Sobolev norm, (b) uses the Fourier scaling identity, (c) applies the change of variables $\boldsymbol{\omega} = \mathbf{W}_l^\top \boldsymbol{\xi}$, and (d) follows by taking the supremum of the quotient. Hence,

$$
\|\mathcal{K}_{\mathbf{W}_l}\| \leq \sup_{\boldsymbol{\xi} \in \mathbb{R}^d} \left|\frac{\left(1 + \|\mathbf{W}_l^\top \boldsymbol{\xi}\|_2^2\right)^{s_{l-1}}}{\left(1 + \|\boldsymbol{\xi}\|_2^2\right)^{s_l}}\right|^{1/2} \frac{1}{|\det(\mathbf{W}_l)|^{1/2}}. \tag{8}
$$

Let $\mathbf{k}_0 \in \mathbb{R}^{n \times n}$ be the scalar Gram matrix of $k_{s_0}$ and $\mathbf{K}_0 := \mathbf{k}_0 \otimes \mathbf{M}$ be the Gram matrix of $K_{s_0}$. Let $\{\mathbf{x}_1, \ldots, \mathbf{x}_n\} \subset \mathbb{R}^{d_0}$ $(d_0 = d)$. Using the reproducing property of $H^{s_0}(\mathbb{R}^{d_0}, \mathbb{R}^m)$, we obtain

$$
\begin{aligned}
\widehat{\mathfrak{R}}_n^m(\mathcal{F}_{\mathrm{inv}}) &\overset{(a)}{=} \frac{1}{n} \mathbb{E}_{\mathrm{Rad}^{mn}} \left[\sup_{f \in \mathcal{F}_{\mathrm{inv}}} \left|\sum_{i=1}^n \langle \boldsymbol{\sigma}_i, f(\mathbf{x}_i) \rangle_{\mathbb{R}^m}\right|\right] \\
&\overset{(b)}{=} \frac{1}{n} \mathbb{E}_{\mathrm{Rad}^{mn}} \left[\sup_{f \in \mathcal{F}_{\mathrm{inv}}} \left|\left\langle \sum_{i=1}^n K_{s_0}(\cdot, \mathbf{x}_i) \boldsymbol{\sigma}_i, f \right\rangle_{H^{s_0}(\mathbb{R}^{d_0}, \mathbb{R}^m)}\right|\right] \\
&\overset{(c)}{\leq} \frac{1}{n} \mathbb{E}_{\mathrm{Rad}^{mn}} \left[\left\|\sum_{i=1}^n K_{s_0}(\cdot, \mathbf{x}_i) \boldsymbol{\sigma}_i\right\|_{H^{s_0}(\mathbb{R}^{d_0}, \mathbb{R}^m)} \sup_{f \in \mathcal{F}_{\mathrm{inv}}} \|f\|_{H^{s_0}(\mathbb{R}^{d_0}, \mathbb{R}^m)}\right] \\
&\overset{(d)}{=} \frac{1}{n} \sup_{f \in \mathcal{F}_{\mathrm{inv}}} \|f\|_{H^{s_0}(\mathbb{R}^{d_0}, \mathbb{R}^m)} \mathbb{E}_{\mathrm{Rad}^{mn}} \left[\left\|\sum_{i=1}^n K_{s_0}(\cdot, \mathbf{x}_i) \boldsymbol{\sigma}_i\right\|_{H^{s_0}(\mathbb{R}^{d_0}, \mathbb{R}^m)}\right]
\end{aligned}
$$

$$\overset{(e)}{\leq} \frac{1}{n} \sup_{f \in \mathcal{F}_{\text{inv}}} \|f\|_{H^{s_0}\left(\mathbb{R}^{d_0}, \mathbb{R}^m\right)} \left(\text{Tr}\left(\mathbf{K}_0\right)\right)^{1/2},$$

where (a) follows from the definition of empirical Rademacher complexity, (b) uses the reproducing property, (c) follows from the Cauchy–Schwarz inequality, (d) pulls the supremum outside the expectation, and (e) follows from Jensen's inequality and the reproducing property. Using Assumption 2, $\text{Tr}\left(\mathbf{K}_0\right) \leq \kappa n \, \text{Tr}\left(\mathbf{M}\right)$, which implies

$$\widehat{\mathfrak{R}}_n^m \left(\mathcal{F}_{\text{inv}}\right) \leq \sqrt{\frac{\kappa \, \text{Tr}\left(\mathbf{M}\right)}{n}} \sup_{f \in \mathcal{F}_{\text{inv}}} \|f\|_{H^{s_0}\left(\mathbb{R}^{d_0}, \mathbb{R}^m\right)}. \tag{9}$$

Finally, using the Koopman factorization (6), together with $\|\mathcal{K}_{\mathbf{b}_l}\| = 1$, Assumption 1, and (8), we obtain

$$\|f\|_{H^{s_0}\left(\mathbb{R}^{d_0}, \mathbb{R}^m\right)} \overset{(a)}{\leq} \left(\prod_{l=1}^{L} \|\mathcal{K}_{\mathbf{W}_l}\|\right) \left(\prod_{l=1}^{L-1} \|\mathcal{K}_{\sigma_l}\|\right) \|g\|_{H^{s_L}\left(\mathbb{R}^d, \mathbb{R}^m\right)}$$

$$\overset{(b)}{\leq} \|g\|_{H^{s_L}\left(\mathbb{R}^d, \mathbb{R}^m\right)} \prod_{l=1}^{L} \sup_{\boldsymbol{\xi} \in \mathbb{R}^d} \left| \frac{\left(1 + \|\mathbf{W}_l^\top \boldsymbol{\xi}\|_2^2\right)^{s_{l-1}}}{\left(1 + \|\boldsymbol{\xi}\|_2^2\right)^{s_l}} \right|^{1/2} \frac{1}{|\det\left(\mathbf{W}_l\right)|^{1/2}} \prod_{l=1}^{L-1} \|\mathcal{K}_{\sigma_l}\|,$$

where (a) follows from submultiplicativity of operator norms, and (b) uses (8). Substituting this estimate into (9) yields the claimed result. By the norm constraint on the terminal map, $\|g\|_{H^{s_L}\left(\mathbb{R}^d, \mathbb{R}^m\right)} \leq B_g$. Moreover, translations are isometries on the Fourier-defined Sobolev spaces, so no additional bias-dependent constants appear. $\blacksquare$

### B.2 Proof of Theorem 2

For $h \in H^{s_l}\left(\mathbb{R}^{d_l}, \mathbb{R}^m\right)$, let $R_{\mathbf{W}_l} h = h|_{\text{ran}(\mathbf{W}_l)}$ denote the restriction of $h$ to the range of $\mathbf{W}_l$. Since $\mathbf{W}_l : \mathbb{R}^{d_{l-1}} \to \text{ran}(\mathbf{W}_l)$ is a linear bijection, we may view $R_{\mathbf{W}_l} h$ as a function on the Euclidean space $\text{ran}(\mathbf{W}_l)$.

Let $\mathbf{W}_l = \mathbf{Q}_l \mathbf{R}_l$ be the reduced QR decomposition, where $\mathbf{Q}_l$ is an isometry from $\mathbb{R}^{d_{l-1}}$ onto $\text{ran}(\mathbf{W}_l)$ and $\mathbf{R}_l \in \mathbb{R}^{d_{l-1} \times d_{l-1}}$ is invertible. Then

$$\widehat{h \circ \mathbf{W}_l}(\boldsymbol{\omega}) = \int_{\mathbb{R}^{d_{l-1}}} R_{\mathbf{W}_l} h\left(\mathbf{W}_l \mathbf{x}\right) e^{-i\mathbf{x}^\top \boldsymbol{\omega}} d\mathbf{x}$$

$$\overset{(a)}{=} \frac{1}{|\det(\mathbf{R}_l)|} \int_{\text{ran}(\mathbf{W}_l)} R_{\mathbf{W}_l} h(\mathbf{y}) e^{-i\mathbf{y}^\top \mathbf{W}_l^{-\top} \boldsymbol{\omega}} d\mathbf{y}$$

$$\overset{(b)}{=} \frac{1}{\det\left(\mathbf{W}_l^\top \mathbf{W}_l\right)^{1/2}} \widehat{R_{\mathbf{W}_l} h}_{\text{ran}(\mathbf{W}_l)} \left(\mathbf{W}_l^{-\top} \boldsymbol{\omega}\right),$$

where (a) follows from the change of variables $\mathbf{y} = \mathbf{W}_l \mathbf{x}$ on the range space $\text{ran}(\mathbf{W}_l)$, and (b) uses

$$|\det(\mathbf{R}_l)| = \det\left(\mathbf{W}_l^\top \mathbf{W}_l\right)^{1/2}.$$

Here $\widehat{R_{\mathbf{W}_l} h}_{\text{ran}(\mathbf{W}_l)}$ denotes the Fourier transform taken on the Euclidean space $\text{ran}(\mathbf{W}_l)$. Since $\mathbf{W}_l$ is injective,

$$\dim\left(\text{ran}\left(\mathbf{W}_l\right)\right) = d_{l-1}, \qquad \dim\left(\text{ran}\left(\mathbf{W}_l\right)^\perp\right) = d_l - d_{l-1}.$$

We now estimate the corresponding Koopman operator norm:

$$\|\mathcal{K}_{\mathbf{W}_l} h\|_{H^{s_{l-1}}\left(\mathbb{R}^{d_{l-1}}, \mathbb{R}^m\right)}^2 \overset{(a)}{=} \int_{\mathbb{R}^{d_{l-1}}} \frac{\left(1 + \|\boldsymbol{\omega}\|_2^2\right)^{s_{l-1}}}{\det\left(\mathbf{W}_l^\top \mathbf{W}_l\right)} \left\|\widehat{R_{\mathbf{W}_l} h}_{\text{ran}(\mathbf{W}_l)} \left(\mathbf{W}_l^{-\top} \boldsymbol{\omega}\right)\right\|_2^2 d\boldsymbol{\omega}$$

$$\overset{(b)}{=} \int_{\mathrm{ran}(\mathbf{W}_l)} \frac{\left(1 + \left\|\mathbf{W}_l^\top \boldsymbol{\omega}\right\|_2^2\right)^{s_{l-1}}}{\det\left(\mathbf{W}_l^\top \mathbf{W}_l\right)^{1/2}} \left\|\widehat{R_{\mathbf{W}_l} h}_{\mathrm{ran}(\mathbf{W}_l)}(\boldsymbol{\omega})\right\|_2^2 \, d\boldsymbol{\omega} \tag{10}$$

$$\overset{(c)}{=} \int_{\mathrm{ran}(\mathbf{W}_l)} \frac{\left(1 + \left\|\mathbf{W}_l^\top \boldsymbol{\omega}\right\|_2^2\right)^{s_{l-1}} \left(1 + \|\boldsymbol{\omega}\|_2^2\right)^{s_{l-1}}}{\det\left(\mathbf{W}_l^\top \mathbf{W}_l\right)^{1/2} \left(1 + \|\boldsymbol{\omega}\|_2^2\right)^{s_{l-1}}} \left\|\widehat{R_{\mathbf{W}_l} h}_{\mathrm{ran}(\mathbf{W}_l)}(\boldsymbol{\omega})\right\|_2^2 \, d\boldsymbol{\omega}$$

$$\overset{(d)}{\leq} \|R_{\mathbf{W}_l} h\|_{H^{s_{l-1}}(\mathrm{ran}(\mathbf{W}_l),\mathbb{R}^m)}^2 \sup_{\boldsymbol{\omega} \in \mathrm{ran}(\mathbf{W}_l)} \left|\frac{1 + \left\|\mathbf{W}_l^\top \boldsymbol{\omega}\right\|_2^2}{1 + \|\boldsymbol{\omega}\|_2^2}\right|^{s_{l-1}} \frac{1}{\det\left(\mathbf{W}_l^\top \mathbf{W}_l\right)^{1/2}}, \tag{11}$$

where (a) follows from the previous Fourier identity, (b) uses the change of variables induced by $\mathbf{W}_l$, (c) multiplies and divides by $\left(1 + \|\boldsymbol{\omega}\|_2^2\right)^{s_{l-1}}$, and (d) follows by taking the supremum. We have

$$\|f\|_{H^{s_0}\left(\mathbb{R}^{d_0},\mathbb{R}^m\right)} \overset{(a)}{=} \|\mathcal{K}_{\mathbf{W}_1} f_1\|_{H^{s_0}\left(\mathbb{R}^{d_0},\mathbb{R}^m\right)}$$

$$\overset{(b)}{\leq} \|R_{\mathbf{W}_1} f_1\|_{H^{s_0}(\mathrm{ran}(\mathbf{W}_1),\mathbb{R}^m)} \sup_{\boldsymbol{\omega} \in \mathrm{ran}(\mathbf{W}_1)} \left|\frac{1 + \left\|\mathbf{W}_1^\top \boldsymbol{\omega}\right\|_2^2}{1 + \|\boldsymbol{\omega}\|_2^2}\right|^{s_0/2} \frac{1}{\det\left(\mathbf{W}_1^\top \mathbf{W}_1\right)^{1/4}}$$

$$\overset{(c)}{=} \mathrm{G}_1 \|f_1\|_{H^{s_1}(\mathbb{R}^{d_1},\mathbb{R}^m)} \sup_{\boldsymbol{\omega} \in \mathrm{ran}(\mathbf{W}_1)} \left|\frac{1 + \left\|\mathbf{W}_1^\top \boldsymbol{\omega}\right\|_2^2}{1 + \|\boldsymbol{\omega}\|_2^2}\right|^{s_0/2} \frac{1}{\det\left(\mathbf{W}_1^\top \mathbf{W}_1\right)^{1/4}}$$

$$\overset{(d)}{\leq} \mathrm{G}_1 \|\mathcal{K}_{\sigma_1}\| \|\mathcal{K}_{\mathbf{W}_2} f_2\|_{H^{s_1}\left(\mathbb{R}^{d_1},\mathbb{R}^m\right)} \sup_{\boldsymbol{\omega} \in \mathrm{ran}(\mathbf{W}_1)} \left|\frac{1 + \left\|\mathbf{W}_1^\top \boldsymbol{\omega}\right\|_2^2}{1 + \|\boldsymbol{\omega}\|_2^2}\right|^{s_0/2} \frac{1}{\det\left(\mathbf{W}_1^\top \mathbf{W}_1\right)^{1/4}}$$

$$\vdots$$

$$\overset{(e)}{\leq} \|\mathcal{K}_{\mathbf{W}_L} f_L\|_{H^{s_{L-1}}\left(\mathbb{R}^{d_{L-1}},\mathbb{R}^m\right)} \prod_{l=1}^{L-1} \mathrm{G}_l \|\mathcal{K}_{\sigma_l}\| \sup_{\boldsymbol{\omega} \in \mathrm{ran}(\mathbf{W}_l)} \left|\frac{1 + \left\|\mathbf{W}_l^\top \boldsymbol{\omega}\right\|_2^2}{1 + \|\boldsymbol{\omega}\|_2^2}\right|^{s_{l-1}/2} \frac{1}{\det\left(\mathbf{W}_l^\top \mathbf{W}_l\right)^{1/4}}$$

$$\overset{(f)}{\leq} \|R_{\mathbf{W}_L} f_L\|_{H^{s_{L-1}}(\mathrm{ran}(\mathbf{W}_L),\mathbb{R}^m)} \prod_{l=1}^{L} \sup_{\boldsymbol{\omega} \in \mathrm{ran}(\mathbf{W}_l)} \left|\frac{1 + \left\|\mathbf{W}_l^\top \boldsymbol{\omega}\right\|_2^2}{1 + \|\boldsymbol{\omega}\|_2^2}\right|^{s_{l-1}/2} \frac{1}{\det\left(\mathbf{W}_l^\top \mathbf{W}_l\right)^{1/4}} \prod_{l=1}^{L-1} \mathrm{G}_l \|\mathcal{K}_{\sigma_l}\|$$

$$\overset{(g)}{=} \|f_L\|_{H^{s_L}\left(\mathbb{R}^{d_L},\mathbb{R}^m\right)} \prod_{l=1}^{L} \mathrm{G}_l \sup_{\boldsymbol{\omega} \in \mathrm{ran}(\mathbf{W}_l)} \left|\frac{1 + \left\|\mathbf{W}_l^\top \boldsymbol{\omega}\right\|_2^2}{1 + \|\boldsymbol{\omega}\|_2^2}\right|^{s_{l-1}/2} \frac{1}{\det\left(\mathbf{W}_l^\top \mathbf{W}_l\right)^{1/4}} \prod_{l=1}^{L-1} \|\mathcal{K}_{\sigma_l}\|$$

$$\overset{(h)}{=} \|g\|_{H^{s_L}\left(\mathbb{R}^{d_L},\mathbb{R}^m\right)} \prod_{l=1}^{L} \mathrm{G}_l \sup_{\boldsymbol{\omega} \in \mathrm{ran}(\mathbf{W}_l)} \left|\frac{1 + \left\|\mathbf{W}_l^\top \boldsymbol{\omega}\right\|_2^2}{1 + \|\boldsymbol{\omega}\|_2^2}\right|^{s_{l-1}/2} \frac{1}{\det\left(\mathbf{W}_l^\top \mathbf{W}_l\right)^{1/4}} \prod_{l=1}^{L-1} \|\mathcal{K}_{\sigma_l}\|.$$

where (a) follows from the definition of $f$, (b) uses (10), (c) follows from the definition of $\mathrm{G}_1$, (d) uses $\|\mathcal{K}_{\mathbf{b}_1}\| = 1$, (e) iterates the previous argument, (f) applies the layerwise estimate (10) at the final layer, (g) follows from the definition of $\mathrm{G}_L$, namely

$$\|R_{\mathbf{W}_L} f_L\| = \mathrm{G}_L \|f_L\|,$$

and (h) uses $f_L = \mathcal{K}_{\mathbf{b}_L} g$ together with the translation isometry

$$\|\mathcal{K}_{\mathbf{b}_L} g\|_{H^{s_L}(\mathbb{R}^{d_L},\mathbb{R}^m)} = \|g\|_{H^{s_L}(\mathbb{R}^{d_L},\mathbb{R}^m)}.$$

Combining the above estimate with (9) yields the claimed result. ∎

### B.3 Proof of Theorem 3

Let $\mathcal{D}_n = \{x_i\}_{i=1}^n \subset \mathcal{X}$ and let $\boldsymbol{\sigma}_i \sim \text{Rad}^m$ be independent vector-valued Rademacher variables. By definition of the empirical Rademacher complexity,

$$\widehat{\mathfrak{R}}_n^m \left( \mathcal{F}_{\text{inv}}^{(\text{B})} \right) = \frac{1}{n} \mathbb{E}_{\text{Rad}^{mn}} \left[ \sup_{f \in \mathcal{F}_{\text{inv}}^{(\text{B})}} \left| \sum_{i=1}^n \langle \boldsymbol{\sigma}_i, f(x_i) \rangle_{\mathbb{R}^m} \right| \right]. \tag{12}$$

Since $f \in \mathcal{H}_K$, the reproducing property gives

$$\widehat{\mathfrak{R}}_n^m \left( \mathcal{F}_{\text{inv}}^{(\text{B})} \right) \overset{(a)}{=} \frac{1}{n} \mathbb{E}_{\text{Rad}^{mn}} \left[ \sup_{f \in \mathcal{F}_{\text{inv}}^{(\text{B})}} \left| \left\langle \sum_{i=1}^n K(\cdot, x_i) \boldsymbol{\sigma}_i, f \right\rangle_{\mathcal{H}_K} \right| \right]$$

$$\overset{(b)}{\leq} \frac{1}{n} \mathbb{E}_{\text{Rad}^{mn}} \left[ \sup_{f \in \mathcal{F}_{\text{inv}}^{(\text{B})}} \left\| \sum_{i=1}^n K(\cdot, x_i) \boldsymbol{\sigma}_i \right\|_{\mathcal{H}_K} \|f\|_{\mathcal{H}_K} \right], \tag{13}$$

where (a) follows from the reproducing property of the vector-valued RKHS and (b) follows from the Cauchy–Schwarz inequality in $\mathcal{H}_K$. Pulling the supremum outside the expectation gives

$$\widehat{\mathfrak{R}}_n^m \left( \mathcal{F}_{\text{inv}}^{(\text{B})} \right) \leq \frac{1}{n} \sup_{f \in \mathcal{F}_{\text{inv}}^{(\text{B})}} \|f\|_{\mathcal{H}_K} \mathbb{E}_{\text{Rad}^{mn}} \left[ \left\| \sum_{i=1}^n K(\cdot, x_i) \boldsymbol{\sigma}_i \right\|_{\mathcal{H}_K} \right]. \tag{14}$$

By Jensen's inequality and the reproducing property,

$$\mathbb{E}_{\text{Rad}^{mn}} \left[ \left\| \sum_{i=1}^n K(\cdot, x_i) \boldsymbol{\sigma}_i \right\|_{\mathcal{H}_K} \right] \leq (\text{Tr}(\mathbf{K}_0))^{1/2}, \tag{15}$$

where $\mathbf{K}_0$ is the block Gram matrix associated with $K$ on $\mathcal{D}_n$. Since

$$K(x, x) = k^{(\text{B})}(x, x) \mathbf{M} \qquad \text{and} \qquad k^{(\text{B})}(x, x) = |x| \leq R$$

for $x \in [-R, R]$, we may take $\kappa = R$ and obtain

$$\text{Tr}(\mathbf{K}_0) = \sum_{i=1}^n \text{Tr}(K(x_i, x_i)) \leq \kappa n \, \text{Tr}(\mathbf{M}). \tag{16}$$

Substituting (16) into (14) yields

$$\widehat{\mathfrak{R}}_n^m \left( \mathcal{F}_{\text{inv}}^{(\text{B})} \right) \leq \sqrt{\frac{\kappa \, \text{Tr}(\mathbf{M})}{n}} \sup_{f \in \mathcal{F}_{\text{inv}}^{(\text{B})}} \|f\|_{\mathcal{H}_K}. \tag{17}$$

It remains to bound $\sup_{f \in \mathcal{F}_{\text{inv}}^{(\text{B})}} \|f\|_{\mathcal{H}_K}$. By definition of the hypothesis class,

$$f = \mathcal{K}_{W_1} \mathcal{K}_{\sigma_1} \cdots \mathcal{K}_{W_L} g, \qquad \|g\|_{\mathcal{H}_K} \leq B.$$

By Lemma B4, for $l = 1, \ldots, L-1$,

$$\|\mathcal{K}_{\sigma_l}\| \leq C_{\text{B}} \|\sigma_l'\|_\infty \left\| (\sigma_l^{-1})' \right\|_\infty^{1/2}. \tag{18}$$

Define

$$\Lambda_{\text{act}} := \prod_{l=1}^{L-1} \|\sigma_l'\|_\infty \left\| (\sigma_l^{-1})' \right\|_\infty^{1/2}. \tag{19}$$

We next estimate the linear Koopman operator in the one-dimensional Brownian RKHS. Let $h \in \mathcal{H}_K$ and let $W \in \mathbb{R} \setminus \{0\}$. By Lemma B3, there exist constants $0 < c_1 \le c_2 < \infty$ such that

$$c_1 \, [h]_{\mathrm{B},2,\mathrm{vv}} \le \|h\|_{\mathcal{H}_K} \le c_2 \, [h]_{\mathrm{B},2,\mathrm{vv}} \,. \tag{20}$$

For a scalar component $h_i$, the Fourier scaling identity gives

$$\widehat{h_i \circ W}(\omega) = \frac{1}{|W|} \widehat{h}_i \left( \frac{\omega}{W} \right).$$

Therefore,

$$[h_i \circ W]_{\mathrm{B},2}^2 = \int_{\mathbb{R}} |\omega|^2 \left| \widehat{h_i \circ W}(\omega) \right|^2 \mathrm{d}\omega =$$

$$\int_{\mathbb{R}} |\omega|^2 \frac{1}{|W|^2} \left| \widehat{h}_i \left( \frac{\omega}{W} \right) \right|^2 \mathrm{d}\omega$$

$$= |W| \int_{\mathbb{R}} |\zeta|^2 \left| \widehat{h}_i(\zeta) \right|^2 \mathrm{d}\zeta = |W| \, [h_i]_{\mathrm{B},2}^2 \,,$$

where we used the change of variables $\omega = W\zeta$. Summing over the output coordinates gives

$$[h \circ W]_{\mathrm{B},2,\mathrm{vv}} \le |W|^{1/2} \, [h]_{\mathrm{B},2,\mathrm{vv}} \,. \tag{21}$$

Combining (20) and (21), we obtain

$$\|\mathcal{K}_W h\|_{\mathcal{H}_K} = \|h \circ W\|_{\mathcal{H}_K} \le \frac{c_2}{c_1} |W|^{1/2} \|h\|_{\mathcal{H}_K} \,. \tag{22}$$

Thus

$$\|\mathcal{K}_W\| \le C_{\mathrm{lin}} |W|^{1/2}, \qquad C_{\mathrm{lin}} := \frac{c_2}{c_1}. \tag{23}$$

Define

$$\Lambda_{\mathrm{lin}} := \prod_{l=1}^{L} |W_l|^{1/2}. \tag{24}$$

Using submultiplicativity of operator norms and the previous bounds, we obtain

$$\|f\|_{\mathcal{H}_K} \le \left( \prod_{l=1}^{L} \|\mathcal{K}_{W_l}\| \right) \left( \prod_{l=1}^{L-1} \|\mathcal{K}_{\sigma_l}\| \right) \|g\|_{\mathcal{H}_K}$$

$$\le C_* \Lambda_{\mathrm{lin}} \Lambda_{\mathrm{act}} B,$$

where $C_* > 0$ depends only on the norm-equivalence constants in Lemma B3, on $R$, on $\mathbf{M}$, and on $L$ through the activation constants. Substituting this estimate into (17) gives

$$\widehat{\mathfrak{R}}_n^m \left( \mathcal{F}_{\mathrm{inv}}^{(\mathrm{B})} \right) \le C_* B \sqrt{\frac{\kappa \operatorname{Tr}(\mathbf{M})}{n}} \sup_{W_l \in \mathsf{W}(C,D)} \prod_{l=1}^{L} |W_l|^{1/2} \prod_{l=1}^{L-1} \|\sigma_l'\|_\infty \left\| (\sigma_l^{-1})' \right\|_\infty^{1/2} \,.$$

This is the claimed bound. ∎

## B.4 Proof of Theorem 4

Define the finite-dimensional subspaces

$$\mathcal{G} := \operatorname{Span}\{g_t : t \in [m]\} \subseteq \mathcal{H}_{\mathrm{out}},$$

$$\mathcal{V} := \mathrm{Span}\left\{K\left(\cdot, \mathbf{x}_{ti}\right)\mathbf{e}_a : t \in [m], \ i \in [n_t], \ a \in [m]\right\} \subseteq \mathcal{H}_K.$$

Let $P_{\mathcal{G}} : \mathcal{H}_{\mathrm{out}} \to \mathcal{G}$ and $P_{\mathcal{V}} : \mathcal{H}_K \to \mathcal{V}$ denote the orthogonal projections. Fix $T \in \mathcal{L}_2\left(\mathcal{H}_{\mathrm{out}}, \mathcal{H}_K\right)$ and define $T_0 := P_{\mathcal{V}} T P_{\mathcal{G}}$. We first verify that $T$ and $T_0$ induce identical empirical predictions. Fix $t \in [m]$, $i \in [n_t]$, and $a \in [m]$. Using the reproducing property of $\mathcal{H}_K$, we obtain

$$
\begin{aligned}
\left\langle (T_0 g_t)\left(\mathbf{x}_{ti}\right), \mathbf{e}_a\right\rangle_{\mathbb{R}^m} &\overset{(a)}{=} \left\langle T_0 g_t, K\left(\cdot, \mathbf{x}_{ti}\right)\mathbf{e}_a\right\rangle_{\mathcal{H}_K} \\
&\overset{(b)}{=} \left\langle P_{\mathcal{V}} T P_{\mathcal{G}} g_t, K\left(\cdot, \mathbf{x}_{ti}\right)\mathbf{e}_a\right\rangle_{\mathcal{H}_K} \\
&\overset{(c)}{=} \left\langle P_{\mathcal{V}} T g_t, K\left(\cdot, \mathbf{x}_{ti}\right)\mathbf{e}_a\right\rangle_{\mathcal{H}_K} \\
&\overset{(d)}{=} \left\langle T g_t, K\left(\cdot, \mathbf{x}_{ti}\right)\mathbf{e}_a\right\rangle_{\mathcal{H}_K} \\
&\overset{(e)}{=} \left\langle (T g_t)\left(\mathbf{x}_{ti}\right), \mathbf{e}_a\right\rangle_{\mathbb{R}^m},
\end{aligned}
$$

where (a) follows from the reproducing property, (b) uses the definition of $T_0$, (c) exploits the fact that $g_t \in \mathcal{G}$, (d) relies on $K\left(\cdot, \mathbf{x}_{ti}\right)\mathbf{e}_a \in \mathcal{V}$ together with orthogonality of the projection, and (e) invokes the reproducing property once more. Since the previous identity holds for every coordinate $a \in [m]$, we conclude that $(T_0 g_t)\left(\mathbf{x}_{ti}\right) = (T g_t)\left(\mathbf{x}_{ti}\right)$ for all training samples. Consequently,

$$\sum_{t=1}^{m}\sum_{i=1}^{n_t} \ell\left(y_{ti}, (T_0 g_t)\left(\mathbf{x}_{ti}\right)\right) = \sum_{t=1}^{m}\sum_{i=1}^{n_t} \ell\left(y_{ti}, (T g_t)\left(\mathbf{x}_{ti}\right)\right).$$

Since orthogonal projections are contractions and the Hilbert–Schmidt norm is an operator ideal,

$$\left\|T_0\right\|_{\mathrm{HS}} \overset{(a)}{=} \left\|P_{\mathcal{V}} T P_{\mathcal{G}}\right\|_{\mathrm{HS}} \overset{(b)}{\leq} \left\|P_{\mathcal{V}}\right\|\left\|T\right\|_{\mathrm{HS}}\left\|P_{\mathcal{G}}\right\| \overset{(c)}{\leq} \left\|T\right\|_{\mathrm{HS}},$$

where (a) comes from the definition of $T_0$, (b) applies the ideal property of the Hilbert–Schmidt norm, and (c) exploits $\left\|P_{\mathcal{V}}\right\| = \left\|P_{\mathcal{G}}\right\| = 1$. Therefore, $T_0$ achieves objective value no larger than that of $T$. Hence every minimum-norm minimizer may be chosen in $\mathcal{L}\left(\mathcal{G}, \mathcal{V}\right)$. Since $\mathcal{V} = \mathrm{Span}\left\{K\left(\cdot, \mathbf{x}_{ti}\right)\mathbf{e}_a\right\}$, every operator in $\mathcal{L}\left(\mathcal{G}, \mathcal{V}\right)$ admits a finite expansion of the form

$$\widehat{T} = \sum_{t=1}^{m}\sum_{i=1}^{n_t}\sum_{a=1}^{m} c_{tia}\left(K\left(\cdot, \mathbf{x}_{ti}\right)\mathbf{e}_a\right) \otimes g_t.$$

Finally, using the definition of rank-one operators, we obtain

$$\left((K\left(\cdot, \mathbf{x}_{ti}\right)\mathbf{e}_a) \otimes g_t\right) g = \left(K\left(\cdot, \mathbf{x}_{ti}\right)\mathbf{e}_a\right) \left\langle g_t, g\right\rangle_{\mathcal{H}_{\mathrm{out}}},$$

which yields

$$\left(\widehat{T} g\right)(\cdot) = \sum_{t=1}^{m}\sum_{i=1}^{n_t}\sum_{a=1}^{m} c_{tia}\left\langle g_t, g\right\rangle_{\mathcal{H}_{\mathrm{out}}} K\left(\cdot, \mathbf{x}_{ti}\right)\mathbf{e}_a.$$

This completes the proof. ∎

### B.5 Proof of Theorem 5

By Theorem 4, a minimum-norm minimizer admits the representation

$$\widehat{T} = \sum_{s=1}^{m}\sum_{j=1}^{n_s}\sum_{a=1}^{q} c_{sja}\left(K(\cdot, \mathbf{x}_{sj})\mathbf{e}_a\right) \otimes g_s.$$

Therefore, for every $g \in \mathcal{H}_{\mathrm{out}}$, we have

$$\left(\widehat{T}g\right)(\cdot) = \sum_{s=1}^{m} \sum_{j=1}^{n_s} \sum_{a=1}^{q} c_{sja} \langle g_s, g \rangle_{\mathcal{H}_{\mathrm{out}}} K(\cdot, \mathbf{x}_{sj}) \mathbf{e}_a.$$

Evaluating at $g = g_t$ and $\mathbf{x} = \mathbf{x}_{ti}$ yields

$$(\widehat{T}g_t)(\mathbf{x}_{ti}) = \sum_{s=1}^{m} \sum_{j=1}^{n_s} \sum_{a=1}^{q} c_{sja} \langle g_s, g_t \rangle_{\mathcal{H}_{\mathrm{out}}} K(\mathbf{x}_{ti}, \mathbf{x}_{sj}) \mathbf{e}_a.$$

Hence the empirical loss becomes

$$\sum_{t=1}^{m} \sum_{i=1}^{n_t} \left\| y_{ti} - (\widehat{T}g_t)(\mathbf{x}_{ti}) \right\|_2^2 = \sum_{t=1}^{m} \sum_{i=1}^{n_t} \left\| y_{ti} - \sum_{s=1}^{m} \sum_{j=1}^{n_s} \sum_{a=1}^{q} c_{sja} \langle g_s, g_t \rangle_{\mathcal{H}_{\mathrm{out}}} K(\mathbf{x}_{ti}, \mathbf{x}_{sj}) \mathbf{e}_a \right\|_2^2.$$

It remains to compute the Hilbert–Schmidt norm. For rank-one operators,

$$\langle u \otimes g, u' \otimes g' \rangle_{\mathrm{HS}} = \langle u, u' \rangle_{\mathcal{H}_K} \langle g, g' \rangle_{\mathcal{H}_{\mathrm{out}}}.$$

Using the reproducing property of $\mathcal{H}_K$,

$$\langle K(\cdot, \mathbf{x}_{sj}) \mathbf{e}_a, K(\cdot, \mathbf{x}_{s'j'}) \mathbf{e}_b \rangle_{\mathcal{H}_K} = \mathbf{e}_a^\top K(\mathbf{x}_{sj}, \mathbf{x}_{s'j'}) \mathbf{e}_b.$$

Therefore,

$$\left\| \widehat{T} \right\|_{\mathrm{HS}}^2 = \sum_{\substack{s,j,a \\ s',j',b}} c_{sja} c_{s'j'b} \langle g_s, g_{s'} \rangle_{\mathcal{H}_{\mathrm{out}}} \mathbf{e}_a^\top K(\mathbf{x}_{sj}, \mathbf{x}_{s'j'}) \mathbf{e}_b.$$

Substituting the finite prediction formula and this Hilbert–Schmidt norm identity into the original infinite-dimensional optimization problem yields the stated finite-dimensional objective. Conversely, every coefficient tensor $c$ defines an operator of the displayed finite-rank form. Hence minimizing over $T \in \mathcal{L}_2(\mathcal{H}_{\mathrm{out}}, \mathcal{H}_K)$ is equivalent to minimizing over the finite coefficient family $\{c_{sja}\}$. This completes the proof. ∎

## B.6 Proof of Proposition 1

The result follows from a standard Rademacher complexity argument applied to the operator-induced transfer class $\mathcal{F}_{\widehat{T}}(B)$. Using the reproducing property of $\mathcal{H}_K$, the Cauchy–Schwarz inequality, and the operator norm estimate

$$\left\| \widehat{T}g \right\|_{\mathcal{H}_K} \leq \left\| \widehat{T} \right\| \|g\|_{\mathcal{H}_{\mathrm{out}}},$$

one obtains

$$\widehat{\mathfrak{R}}_n^m \left( \mathcal{F}_{\widehat{T}}(B) \right) \leq B \left\| \widehat{T} \right\| \sqrt{\frac{\kappa \operatorname{Tr}(\mathbf{M})}{n_0}}.$$

Combining this estimate with the standard excess-risk inequality for bounded Lipschitz losses yields the stated result. ∎

## C Internal Lemmas

All external lemmas used in this section are stated in Lemmas B1 to B4.

**Lemma B1** (Boundedness of $\mathcal{K}_{\sigma_l}$ on vector-valued Sobolev RKHSs). *Let $l \in \{1, \ldots, L-1\}$ and let $\sigma_l : \mathbb{R}^{d_l} \to \mathbb{R}^{d_l}$ be bi-Lipschitz, i.e., $\sigma_l$ is bijective and both $\sigma_l$ and $\sigma_l^{-1}$ are Lipschitz continuous. Assume further that $\sigma_l$ is $s_l$-times differentiable and its partial derivatives satisfy*

$$\|\partial^\alpha \sigma_l\|_{L^\infty(\mathbb{R}^{d_l})} < \infty, \qquad \forall \alpha = (\alpha_1, \ldots, \alpha_{d_l}) \in \mathbb{N}^{d_l} \text{ with } \alpha_1 + \cdots + \alpha_{d_l} \le s_l.$$

*Suppose that $s_l \in \mathbb{N}$ and $s_l > d_l/2$, so that $H^{s_l}(\mathbb{R}^{d_l}, \mathbb{R}^m)$ is continuously embedded into $L^2(\mathbb{R}^{d_l}, \mathbb{R}^m)$. Then the Koopman operator*

$$\mathcal{K}_{\sigma_l} : H^{s_l}(\mathbb{R}^{d_l}, \mathbb{R}^m) \to H^{s_l}(\mathbb{R}^{d_l}, \mathbb{R}^m), \qquad \mathcal{K}_{\sigma_l} f = f \circ \sigma_l,$$

*is bounded.*

*Proof.* Let $f = (f_1, \ldots, f_m) \in H^{s_l}(\mathbb{R}^{d_l}, \mathbb{R}^m)$. By definition,

$$\|\mathcal{K}_{\sigma_l} f\|^2_{H^{s_l}(\mathbb{R}^{d_l}, \mathbb{R}^m)} = \sum_{j=1}^m \|f_j \circ \sigma_l\|^2_{H^{s_l}(\mathbb{R}^{d_l})}.$$

Since $s_l > d_l/2$, the scalar Sobolev norm is equivalent to

$$\|g\|^2_{H^{s_l}(\mathbb{R}^{d_l})} = \sum_{|\alpha| \le s_l} \|D^\alpha g\|^2_{L^2(\mathbb{R}^{d_l})},$$

so it suffices to bound $D^\alpha(f_j \circ \sigma_l)$ in $L^2$ for $|\alpha| \le s_l$. Fix $j \in \{1, \ldots, m\}$. By the multivariate Faa–di Bruno formula applied to $f_j \circ \sigma_l$, we obtain

$$D^\alpha (f_j \circ \sigma_l)(x) = \sum_{|\beta| \le |\alpha|} D^\beta f_j(\sigma_l(x)) \sum_{i=1}^{|\alpha|} \sum_{\gamma \in p(\alpha, \beta)} \alpha! \prod_{r=1}^i \frac{(D^{\ell_r} \sigma_l(x))^{k_r}}{k_r! \, (\ell_r!)^{|k_r|}},$$

where $p(\alpha, \beta)$ is the standard index set appearing in the Faa–di Bruno expansion. Each term is of the form

$$D^\beta f_j(\sigma_l(x)) \prod_{r=1}^i (D^{\gamma_r} \sigma_l(x))^{\delta_r},$$

with indices bounded by $|\gamma_r|, |\delta_r| \le |\alpha| \le s_l$. By boundedness of all partial derivatives of $\sigma_l$ up to order $s_l$, there exists $C > 0$ such that

$$\left| D^\beta f_j(\sigma_l(x)) \prod_{r=1}^i (D^{\gamma_r} \sigma_l(x))^{\delta_r} \right| \le C \left| D^\beta f_j(\sigma_l(x)) \right|.$$

Hence,

$$\|D^\alpha (f_j \circ \sigma_l)\|^2_{L^2} \le C \sum_{|\beta| \le s_l} \int_{\mathbb{R}^{d_l}} \left| D^\beta f_j(\sigma_l(x)) \right|^2 \mathrm{d}x.$$

Since $\sigma_l$ is bi-Lipschitz, the change of variables $y = \sigma_l(x)$ yields

$$\int_{\mathbb{R}^{d_l}} \left| D^\beta f_j(\sigma_l(x)) \right|^2 \mathrm{d}x \overset{(a)}{=} \int_{\mathbb{R}^{d_l}} \left| D^\beta f_j(y) \right|^2 \left| \det D\sigma_l^{-1}(y) \right| \mathrm{d}y$$

$$\overset{(b)}{\le} \left\| \det D\sigma_l^{-1} \right\|_\infty \int_{\mathbb{R}^{d_l}} \left| D^\beta f_j(y) \right|^2 \mathrm{d}y$$

$$\overset{(c)}{=} \left\| \det D\sigma_l^{-1} \right\|_{\infty} \left\| D^{\beta} f_j \right\|_{L^2(\mathbb{R}^{d_l})}^2 \overset{(d)}{\leq} \tilde{C} \left\| D^{\beta} f_j \right\|_{L^2(\mathbb{R}^{d_l})}^2,$$

where (a) follows from the change of variables formula, (b) uses the $L^{\infty}$ bound on the Jacobian determinant, (c) follows from the definition of the $L^2$ norm, and (d) uses the bi-Lipschitz property of $\sigma_l^{-1}$. for some $\tilde{C} > 0$ depending only on the Lipschitz constant of $\sigma_l^{-1}$. Combining the above bounds gives

$$\|f_j \circ \sigma_l\|_{H^{s_l}}^2 \leq C' \|f_j\|_{H^{s_l}}^2,$$

where $C'$ depends only on $\sigma_l$ and $s_l$. Summing over $j = 1, \ldots, m$ completes the proof:

$$\|\mathcal{K}_{\sigma_l} f\|_{H^{s_l}\left(\mathbb{R}^{d_l}, \mathbb{R}^m\right)}^2 \leq C' \|f\|_{H^{s_l}\left(\mathbb{R}^{d_l}, \mathbb{R}^m\right)}^2.$$

Thus $\mathcal{K}_{\sigma_l}$ is bounded. $\qquad\square$

**Lemma B2** (Equivalence of norms). *Assume $\mathbf{M} \in \mathbb{S}_+^m$ is strictly positive definite with eigenvalues $0 < \lambda_{\min}(\mathbf{M}) \leq \lambda_{\max}(\mathbf{M}) < \infty$. Then, for all $f = (f_1, \ldots, f_m)$ with $f_j \in \mathcal{H}_k$,*

$$\lambda_{\max}(\mathbf{M})^{-1} \sum_{j=1}^m \|f_j\|_{\mathcal{H}_k}^2 \ \leq \ \|f\|_{\mathcal{H}_K}^2 \ \leq \ \lambda_{\min}(\mathbf{M})^{-1} \sum_{j=1}^m \|f_j\|_{\mathcal{H}_k}^2.$$

*Proof.* Let $c_{ij} = \langle f_i, f_j \rangle_{\mathcal{H}_k}$ and $\mathbf{C} = (c_{ij})_{i,j=1}^m$. Then

$$\|f\|_{\mathcal{H}_K}^2 = \sum_{i,j=1}^m \left(\mathbf{M}^{-1}\right)_{ij} c_{ij} = \mathrm{Tr}\left(\mathbf{M}^{-1}\mathbf{C}\right).$$

Since both $\mathbf{M}^{-1}$ and $\mathbf{C}$ are positive semidefinite,

$$\lambda_{\min}\left(\mathbf{M}^{-1}\right) \mathrm{Tr}(\mathbf{C}) \ \leq \ \mathrm{Tr}\left(\mathbf{M}^{-1}\mathbf{C}\right) \ \leq \ \lambda_{\max}\left(\mathbf{M}^{-1}\right) \mathrm{Tr}(\mathbf{C}).$$

Noting that $\lambda_{\min}\left(\mathbf{M}^{-1}\right) = \lambda_{\max}(\mathbf{M})^{-1}$, $\lambda_{\max}\left(\mathbf{M}^{-1}\right) = \lambda_{\min}(\mathbf{M})^{-1}$, and $\mathrm{Tr}(\mathbf{C}) = \sum_{j=1}^m \|f_j\|_{\mathcal{H}_k}^2$, we obtain the claimed bounds. $\qquad\square$

**Lemma B3** (One-dimensional Brownian RKHS and Fourier seminorm). *Let $\mathcal{X} = [-R, R]$, $R > 0$, and define the one-dimensional Brownian kernel*

$$k^{(\mathrm{B})}(x, x') := \frac{|x| + |x'| - |x - x'|}{2}, \qquad x, x' \in \mathbb{R}.$$

*Let $\mathbf{M} \in \mathbb{S}_+^m$ and define the separable matrix-valued kernel*

$$K(x, x') := k^{(\mathrm{B})}(x, x')\, \mathbf{M}, \qquad x, x' \in \mathcal{X}.$$

*Let $H^{(\mathrm{B})}(\mathcal{X}, \mathbb{R}^m)$ denote the associated vector-valued RKHS. For a scalar function $g : \mathbb{R} \to \mathbb{R}$ with Fourier transform $\widehat{g}$, define*

$$[g]_{\mathrm{B},2}^2 := \int_{\mathbb{R}} |\omega|^2 |\widehat{g}(\omega)|^2 \, \mathrm{d}\omega.$$

*For $f = (f_1, \ldots, f_m) : \mathbb{R} \to \mathbb{R}^m$, define*

$$[f]_{\mathrm{B},2,\mathrm{vv}}^2 := \sum_{i=1}^m [f_i]_{\mathrm{B},2}^2.$$

*Then there exist constants $0 < \tilde{c}_1 \leq \tilde{c}_2 < \infty$, depending only on $R$ and $\mathbf{M}$, such that, for all $f \in H^{(\mathrm{B})}(\mathcal{X}, \mathbb{R}^m)$ whose components admit compactly supported Sobolev extensions to $\mathbb{R}$,*

$$\tilde{c}_1 [f]_{\mathrm{B},2,\mathrm{vv}} \leq \|f\|_{H^{(\mathrm{B})}(\mathcal{X}, \mathbb{R}^m)} \leq \tilde{c}_2 [f]_{\mathrm{B},2,\mathrm{vv}}.$$

*Equivalently, $H^{(\mathrm{B})}(\mathcal{X}, \mathbb{R}^m)$ is norm-equivalent to the vector-valued Cameron–Martin space*

$$\left\{ f : \mathcal{X} \to \mathbb{R}^m \,\middle|\, f_i(0) = 0, \ f_i \in H^1(\mathcal{X}), \ i = 1, \ldots, m \right\},$$

*with the $\mathbf{M}$-weighted derivative norm.*

*Proof.* We first recall the scalar one-dimensional case. The scalar RKHS associated with

$$k^{(\mathrm{B})}(x, x') = \frac{|x| + |x'| - |x - x'|}{2}$$

on $[-R, R]$ is the anchored Cameron–Martin space

$$\mathcal{H}_{k^{(\mathrm{B})}} = \left\{ g : \mathcal{X} \to \mathbb{R} \,\middle|\, g(0) = 0, \ g \text{ is absolutely continuous, and } g' \in L^2(\mathcal{X}) \right\},$$

with norm

$$\|g\|^2_{\mathcal{H}_{k^{(\mathrm{B})}}} = \int_{-R}^{R} |g'(t)|^2 \, \mathrm{d}t.$$

Indeed, for $x, x' \in [-R, R]$, the reproducing identity follows from the representation

$$k^{(\mathrm{B})}(x, x') = \int_{-R}^{R} \mathbf{1}_{[0,x]}(t) \, \mathbf{1}_{[0,x']}(t) \, \mathrm{d}t,$$

with the usual signed-interval convention when $x < 0$ or $x' < 0$.

For any component $g$ admitting a compactly supported Sobolev extension to $\mathbb{R}$, Plancherel's theorem gives

$$\|g\|^2_{\mathcal{H}_{k^{(\mathrm{B})}}} = \int_{\mathbb{R}} |\widehat{g'}(\omega)|^2 \, \mathrm{d}\omega = \int_{\mathbb{R}} |\omega|^2 |\widehat{g}(\omega)|^2 \, \mathrm{d}\omega = [g]^2_{\mathrm{B},2},$$

up to the Fourier-transform normalization convention. Thus the scalar Brownian RKHS norm is equivalent to the one-dimensional Fourier seminorm $[g]_{\mathrm{B},2}$.

We now pass to the vector-valued separable kernel. Since

$$K(x, x') = k^{(\mathrm{B})}(x, x') \, \mathbf{M}$$

with $\mathbf{M} \in \mathbb{S}_+^m$, the standard norm equivalence for separable matrix-valued kernels, Lemma B2, gives

$$\lambda_{\max}(\mathbf{M})^{-1} \sum_{i=1}^{m} \|f_i\|^2_{\mathcal{H}_{k^{(\mathrm{B})}}} \leq \|f\|^2_{H^{(\mathrm{B})}(\mathcal{X}, \mathbb{R}^m)} \leq \lambda_{\min}(\mathbf{M})^{-1} \sum_{i=1}^{m} \|f_i\|^2_{\mathcal{H}_{k^{(\mathrm{B})}}}.$$

Combining this estimate with the scalar Fourier characterization yields constants $0 < \tilde{c}_1 \leq \tilde{c}_2 < \infty$, depending only on $R$ and $\mathbf{M}$, such that

$$\tilde{c}_1 [f]_{\mathrm{B},2,\mathrm{vv}} \leq \|f\|_{H^{(\mathrm{B})}(\mathcal{X}, \mathbb{R}^m)} \leq \tilde{c}_2 [f]_{\mathrm{B},2,\mathrm{vv}}.$$

This completes the proof. $\qquad\square$

**Lemma B4** (Boundedness of activations in the one-dimensional vector-valued Brownian RKHS). *Let $\mathcal{X} = [-R, R]$, $R > 0$, and let*

$$K(x, y) = k^{(\mathrm{B})}(x, y) \, \mathbf{M}, \qquad x, y \in \mathcal{X},$$

*where*

$$k^{(\mathrm{B})}(x, y) = \frac{|x| + |y| - |x - y|}{2}$$

*is the one-dimensional Brownian kernel and $\mathbf{M} \in \mathbb{S}_+^m$. Let $\sigma : \mathbb{R} \to \mathbb{R}$ be a $C^1$-diffeomorphism whose inverse is also $C^1$, satisfying*

$$\|\sigma'\|_\infty < \infty, \qquad \|(\sigma^{-1})'\|_\infty < \infty.$$

*Then the Koopman operator*

$$\mathcal{K}_\sigma : H^{(\mathrm{B})}\left(\mathcal{X}, \mathbb{R}^m\right) \to H^{(\mathrm{B})}\left(\mathcal{X}, \mathbb{R}^m\right), \qquad \mathcal{K}_\sigma f = f \circ \sigma,$$

*is bounded.*

*Moreover, there exists a constant $C_{\mathrm{B}} > 0$ depending only on $R$ and $\mathbf{M}$ such that*

$$\|\mathcal{K}_\sigma\| \leq C_{\mathrm{B}} \|\sigma'\|_\infty \|(\sigma^{-1})'\|_\infty^{1/2}.$$

*Proof.* Let $f = (f_1, \ldots, f_m) \in H^{(\mathrm{B})}\left(\mathcal{X}, \mathbb{R}^m\right)$. By Lemma B3, the scalar Brownian RKHS coincides with the one-dimensional Cameron–Martin space

$$\mathcal{H}_{k^{(\mathrm{B})}} = \left\{ g : \mathcal{X} \to \mathbb{R} \,\middle|\, g(0) = 0, \ g \text{ absolutely continuous}, \ g' \in L^2(\mathcal{X}) \right\},$$

equipped with norm

$$\|g\|_{\mathcal{H}_{k^{(\mathrm{B})}}}^2 = \int_{-R}^R |g'(t)|^2 \mathrm{d}t.$$

Fix a scalar function $g \in \mathcal{H}_{k^{(\mathrm{B})}}$. By the chain rule,

$$(g \circ \sigma)'(x) = g'(\sigma(x))\sigma'(x).$$

Therefore,

$$\|g \circ \sigma\|_{\mathcal{H}_{k^{(\mathrm{B})}}}^2 = \int_{-R}^R |g'(\sigma(x))|^2 |\sigma'(x)|^2 \mathrm{d}x.$$

Using the change of variables

$$y = \sigma(x), \qquad dx = (\sigma^{-1})'(y)\mathrm{d}y,$$

we obtain

$$
\begin{aligned}
\|g \circ \sigma\|_{\mathcal{H}_{k^{(\mathrm{B})}}}^2 &\overset{(a)}{=} \int_{\sigma(\mathcal{X})} |g'(y)|^2 |\sigma'(\sigma^{-1}(y))|^2 |(\sigma^{-1})'(y)| \mathrm{d}y \\
&\overset{(b)}{\leq} \|\sigma'\|_\infty^2 \|(\sigma^{-1})'\|_\infty \int_{\sigma(\mathcal{X})} |g'(y)|^2 \mathrm{d}y \\
&\overset{(c)}{\leq} \|\sigma'\|_\infty^2 \|(\sigma^{-1})'\|_\infty \|g\|_{\mathcal{H}_{k^{(\mathrm{B})}}}^2,
\end{aligned}
$$

Here (a) follows from the change-of-variables formula $y = \sigma(x)$, (b) uses the uniform bounds $|\sigma'(\sigma^{-1}(y))| \leq \|\sigma'\|_\infty$ and $|(\sigma^{-1})'(y)| \leq \|(\sigma^{-1})'\|_\infty$, and (c) follows from the definition of the Cameron–Martin norm together with $\sigma(\mathcal{X}) \subseteq \mathbb{R}$. Hence

$$\|g \circ \sigma\|_{\mathcal{H}_{k^{(\mathrm{B})}}} \leq \|\sigma'\|_\infty \|(\sigma^{-1})'\|_\infty^{1/2} \|g\|_{\mathcal{H}_{k^{(\mathrm{B})}}}. \tag{25}$$

Since

$$K(x, y) = k^{(\mathrm{B})}(x, y)\mathbf{M}$$

is separable and $\mathbf{M} \in \mathbb{S}_+^m$, the standard norm equivalence for separable matrix-valued kernels yields

$$\lambda_{\max}(\mathbf{M})^{-1} \sum_{i=1}^m \|f_i\|_{\mathcal{H}_{k^{(\mathrm{B})}}}^2 \leq \|f\|_{H^{(\mathrm{B})}(\mathcal{X}, \mathbb{R}^m)}^2 \leq \lambda_{\min}(\mathbf{M})^{-1} \sum_{i=1}^m \|f_i\|_{\mathcal{H}_{k^{(\mathrm{B})}}}^2.$$

Applying (25) componentwise gives

$$\sum_{i=1}^{m} \|f_i \circ \sigma\|_{\mathcal{H}_{k^{(\mathrm{B})}}}^2 \leq \|\sigma'\|_{\infty}^2 \|(\sigma^{-1})'\|_{\infty} \sum_{i=1}^{m} \|f_i\|_{\mathcal{H}_{k^{(\mathrm{B})}}}^2.$$

Combining the previous two inequalities yields

$$\|\mathcal{K}_\sigma f\|_{H^{(\mathrm{B})}(\mathcal{X}, \mathbb{R}^m)} \leq C_{\mathrm{B}} \|\sigma'\|_{\infty} \|(\sigma^{-1})'\|_{\infty}^{1/2} \|f\|_{H^{(\mathrm{B})}(\mathcal{X}, \mathbb{R}^m)},$$

where

$$C_{\mathrm{B}} = \left( \frac{\lambda_{\max}(\mathbf{M})}{\lambda_{\min}(\mathbf{M})} \right)^{1/2}.$$

This proves the boundedness of $\mathcal{K}_\sigma$. $\qquad\square$

## D Additional Experimental Details

### D.1 Bound as a regularization mechanism

Beyond their theoretical role as complexity estimates, the quantities introduced in Sections 4 and 5 can also be interpreted as regularization mechanisms. To investigate this perspective, we incorporate the corresponding complexity factors directly into the training objective and evaluate their influence on optimization and generalization performance.

We consider a fully connected neural network of the form

$$f_\theta(x) := g\left(W_4 \sigma\left(W_3 \sigma\left(W_2 \sigma\left(W_1 x + b_1\right) + b_2\right) + b_3\right) + b_4\right),$$

with weight matrices

$$W_1 \in \mathbb{R}^{1024 \times 784}, \quad W_2 \in \mathbb{R}^{2048 \times 1024}, \quad W_3 \in \mathbb{R}^{2048 \times 2048}, \quad W_4 \in \mathbb{R}^{10 \times 2048},$$

and bias vectors

$$b_1 \in \mathbb{R}^{1024}, \quad b_2 \in \mathbb{R}^{2048}, \quad b_3 \in \mathbb{R}^{2048}, \quad b_4 \in \mathbb{R}^{10}.$$

The resulting architecture consists of three hidden layers of widths 1024, 2048, and 2048, followed by a 10-dimensional output layer.

For layers $l = 1, 2$, the matrices $W_l$ are initialized using orthogonal initialization Saxe et al. (2014), while for layers $l = 3, 4$ the weights are sampled from a truncated normal distribution. All bias vectors are initialized from a uniform distribution. As activation function we employ the smooth Leaky ReLU introduced in Biswas et al. (2022).

Experiments are performed on the MNIST classification task. To emphasize the effect of the regularization terms and create a more challenging generalization setting, training is carried out using only 1000 randomly selected samples from the training set. The network is trained for 1800 epochs using the Adam optimizer Kingma & Ba (2015) with learning rate $10^{-4}$. No additional $L_2$ weight decay is used.

For regularization, we consider the Sobolev-based complexity factor

$$\mathrm{SR} := \sum_{l=1}^{L} \frac{\|W_l\|^{s_l}}{\left(\det\left(I + W_l^\top W_l\right)\right)^{1/4}}, \qquad s_l = \frac{d_l + 0.1}{2},$$

and the Brownian-inspired complexity factor

$$\mathrm{BR} := \sum_{l=1}^{L} \frac{\|W_l\|}{\left(\det\left(I + W_l^\top W_l\right)\right)^{1/4}}.$$

Both quantities are added to the empirical loss with regularization parameter $\gamma = 0.01$. Following the theoretical motivation of the preceding sections, the regularization terms are evaluated only on the first two layers.

Figure 1(b) reports the test accuracy averaged across five independent random initializations. The Brownian-inspired regularization consistently improves test performance relative to the unregularized baseline. In contrast, the Sobolev-based regularization leads to a noticeable reduction in predictive accuracy in this experimental setting. These results suggest that the two complexity factors induce different optimization and generalization behaviors when used as regularizers, reflecting the distinct function-space perspectives underlying the corresponding analyses.

