# OpenReview forum: "Operator-Based Generalization Bounds for Multitask Deep Learning"
_TMLR — Under review for TMLR_

### Review · Reviewer_MAN9 · 2026-06-09

**Summary Of Contributions:**

The paper proposes Koopman-operator generalization bounds for deep multi-output networks in vector-valued RKHSs. It extends a recent scalar Sobolev-RKHS Koopman analysis to separable matrix-valued kernels, introduces a Brownian RKHS variant intended to remove Sobolev smoothness exponents, and adds a shared-operator learning formulation with a representer theorem and transfer bound.

## Strengths
- The paper addresses an interesting question: how Koopman-based generalization bounds change when moving from scalar Sobolev RKHSs to vector-valued and Brownian-type function spaces.
- The vector-valued Rademacher-complexity setup is natural for multi-output prediction and makes the dependence on the output kernel trace explicit.
- The shared-operator learning section is a potentially useful bridge between Koopman composition operators and multitask transfer.

## Weaknesses
- Several core mathematical steps appear incorrect or under-specified, especially the multidimensional Brownian RKHS characterization, translation boundedness in the Brownian RKHS, the injective-layer Fourier argument, and the determinant/shape conventions.
- Some stated bounds are missing constants or depth dependence, and some hypothesis classes are unbounded because the terminal map g is not norm-constrained.
- The novelty over existing vector-valued RKHS, multitask kernel, and Koopman representer-theorem work is not clearly separated.
- The experiments do not convincingly validate the claimed bounds: the plotted “bounds” omit major theoretical factors, use stabilized determinants not derived in full, and are tested mostly on a single multiclass MNIST setup rather than a multitask transfer setting.

**Audience:**

Yes

**Audience Explanation:**

The topic is relevant to TMLR’s audience. Operator-theoretic generalization, full-rank weight matrices, vector-valued RKHSs, and multitask transfer are all active areas of interest. The central idea of comparing Koopman bounds across Sobolev and Brownian-type function spaces could be useful if the mathematical formulation is repaired.

**Claims And Evidence:**

No

**Claims Explanation:**

The main concern is technical soundness. The Brownian RKHS argument is currently not convincing. The paper defines a one-dimensional Brownian kernel and later uses a multidimensional version, but the multidimensional RKHS is not rigorously specified. In Lemma B3, the kernel appears to be an additive sum of one-dimensional Brownian kernels, yet the proof treats the resulting RKHS as equivalent to a full first-order Sobolev space on $[−R,R]^d$. For an additive kernel, the RKHS is an additive function space, not all of $H^1$ on the cube. Standard Brownian RKHS results identify the one-dimensional Brownian RKHS with absolutely continuous functions rooted at zero and square-integrable derivative, so the multidimensional extension requires much more care [9].

The Brownian translation argument also appears wrong as stated. The Brownian RKHS associated with the standard Brownian covariance is rooted at the origin, so translating the input does not generally preserve the condition $f(0)=0$. The proof states that translation operators have norm one by “translation invariance of the Brownian Fourier seminorm,” but the Brownian kernel itself is not translation invariant. This affects Theorem 3 because the network factorization includes bias shifts at every layer.

The injective-layer Sobolev proof is not convincing. For an injective map $W: R^{d_{l-1}} -> R^{d_l}$, the Fourier transform of $h(Wx)$ involves the Fourier transform of the restriction of h to $ran(W)$, or equivalently an integration over orthogonal directions, not simply evaluation of the full Fourier transform $\hat h$ on $ran(W)$. The current proof therefore seems to miss a trace/restriction argument and the needed regularity assumptions.

There are also several internal inconsistencies. The definition of $W_l(C,D)$ uses a matrix shape inconsistent with the stated map direction; with the written shape, $det(W^T W)$ would be singular when $d_l >= d_{l-1}$. Corollary 1 appears to drop depth dependence: applying Theorem 1 layerwise should yield a product over layers, e.g. factors like $max{1,C^s}^L$ and $D^{-L/2}$ under the paper’s own per-layer constraints, not a single $max{1,C^s}$ and $D^{-1/2}$. Theorem 3 defines a class with $g ∈ H^{(B)}$ but gives a bound proportional to $||g||$; without fixing $g$ or imposing $||g|| ≤ B$, the Rademacher complexity is infinite.

**Requested Changes:**

## Critical
- Correct the multidimensional Brownian RKHS definition and prove the exact RKHS norm used. The paper must specify whether the kernel is additive, tensor-product, Lévy Brownian, or another construction, and the claimed equivalence to a Sobolev seminorm must match that kernel.
- Fix the Brownian translation and bias-shift analysis. If the standard Brownian RKHS is rooted at zero, translations are not isometries. The authors should either use a Brownian-motion-started-at-random RKHS, add constants to the space, restrict biases, or prove the correct boundedness result.
- Redo the injective-layer Koopman proof. The Fourier transform of a restriction to $ran(W)$ must be handled using a valid trace/restriction theorem, with all required Sobolev regularity assumptions and constants.
- Fix the matrix shape and determinant conventions in Theorem 2. The current definition of $W_l(C,D)$ conflicts with the stated injective map direction and makes $det(W^T W)$ problematic.
- Add norm constraints or fixed-terminal-map assumptions to all hypothesis classes. In particular, Theorem 3 is unbounded as written if g ranges over the whole Brownian RKHS.
- Restore all missing constants and depth factors. The Brownian theorem statement omits constants introduced in the proof, and Corollary 1 appears to lose the product over layers.
## Would strengthen
- Add missing citations to the core vector-valued RKHS and multitask-kernel literature, especially [3–5], and to Brownian/Cameron–Martin RKHS references such as [9].
- Add a limitations section explaining that the current theory requires injective or regularized full-rank layers, smooth invertible activations, compact support or domain-control assumptions, and carefully chosen terminal maps.
- Improve notation consistency. The paper repeatedly conflates m output dimension with M number of tasks, duplicates sentences in Section 6, and states that “all proofs appear in the main text” while placing them in the appendix.

## References

[1] Koopman-Based Generalization Bound: New Aspect for Full-Rank Weights. Hashimoto et al. 2024.

[2] Representer Theorem for Learning Koopman Operators. Khosravi. 2023.

[3] Learning Multiple Tasks with Kernel Methods. Evgeniou et al. 2005.

[4] On Learning Vector-Valued Functions. Micchelli and Pontil. 2005.

[5] Universal Multi-Task Kernels. Caponnetto et al. 2008.

[6] Bounds for Linear Multi-Task Learning. Maurer. 2006.

[7] Excess Risk Bounds for Multitask Learning with Trace Norm Regularization. Maurer and Pontil. 2013.

[8] The Benefit of Multitask Representation Learning. Maurer et al. 2016.

[9] Reproducing Kernel Hilbert Spaces of Gaussian Priors. van der Vaart and van Zanten. 2008.

---

> ### Author Response · Authors · 2026-06-11
> **Author Response and Planned Revision**
>
> We thank the reviewers for the detailed and thoughtful reports. We appreciate the careful technical reading, the extensive comments, and the references provided. We also appreciate the reviewers' engagement with both the theoretical and empirical aspects of the manuscript.
>
> A central point raised by the reviews concerns the Brownian section and, in particular, the relationship between the Brownian kernel construction, the associated RKHS, and the first-order Fourier/Cameron--Martin-type energy used in the analysis.
>
> In the current manuscript, the intention was to capture a first-order Brownian/Cameron--Martin type regularity through a Fourier-energy perspective and to compare the resulting Koopman-based bounds with Sobolev-type constructions. We agree that the precise relationship between the Fourier-energy characterization used in the analysis, the chosen Brownian kernel construction, and the associated RKHS should be formulated more carefully and justified more rigorously.
>
> In the revision we will explicitly clarify the kernel construction being used and provide a precise characterization of the corresponding function space. In particular, we will carefully distinguish between:
>
> 1. one-dimensional Brownian/Cameron--Martin RKHS constructions;
>
> 2. multidimensional Brownian-type kernel constructions (e.g., additive, tensor-product/Brownian-sheet, or integral/ridge constructions);
>
> 3. the first-order Fourier/Dirichlet energy $[f]_{B,2}$ and its role in the analysis.
>
> Accordingly, the revised manuscript will remove ambiguity between the kernel definition, the associated RKHS, and the Fourier/Sobolev-type seminorm used in the proofs.
>
> In addition, we will:
>
> * revise the Brownian translation/bias analysis and make the assumptions explicit;
> * revise the injective-layer Fourier argument and clarify the restriction/trace component;
> * correct the matrix-shape and determinant conventions;
> * add the missing norm constraints (or equivalent fixed-terminal-map assumptions) in Theorem 3;
> * restore omitted constants and depth-dependent factors;
> * expand the discussion of assumptions and limitations;
> * incorporate the suggested references and improve notation consistency;
> * revise the presentation of the experimental section to better align the empirical claims with the scope of the theoretical results.
>
> We found the reviews extremely helpful and are currently preparing a detailed revision together with a point-by-point response addressing each comment individually.

---

### Review · Reviewer_2X7W · 2026-06-10

**Summary Of Contributions:**

The paper proposes an operator-theoretic framework to analyze generalization in DNNs and multitask learning settings.

### Strengths:
- The paper succeeds in providing a framework, including RKHS learning theory and Koopman-style operator learning, into a unified language. The extension of Koopman analysis to vvRKHSs is also an interesting contribution.

### Weaknesses:
- Sobolev and Brownian RKHS regimes are fundamentally different, due to the change in hypothesis space and smoothness assumptions. Because the RKHS geometry are altered entirely, this seems like model substitution rather than a controlled ablation. As a result, the claim that the "Brownian formulation is better" throughout Section 1 and 6 does not seem valid.
- There is also no evidence that the authors provide that shows that Brownian RKHS better matches NNs. The authors should show some justification for the same.
- From Section 3, the authors state several smoothness assumptions, such as smooth activations, bounded RKHS embeddings and others, that seems to not apply to several realistic NNs.
- In Section 5, the Sobolev formulation intentionally includes higher-order norms, whereas the Brownian formulation relies on first-order norms. Thus, the observed stability of the Brownian bound during training is expected and it is mathematically guaranteed to grow slower. The advantage gained from being defined as Brownian seems to be overestimated as a result.
- Could the authors provide some actual measures for true generalization gaps, Rademacher complexity, or empirical risk bound tightness? The synthetic experiment described also seems to be very smooth and biased (described below), there is not enough evidence to prove that the Brownian definition is required here.
- Could the authors also validate this on other datasets? Validating these spectral geometry differences on MNIST does not seem sufficient due to its low complexity and strong inductive bias for near-linear classifiers.
- The paper explicitly claims that the Brownian bound is tighter and that its regularization improves performance. However, the actual evidence demonstrates that the Brownian mathematical expression grows slower. The conclusions far exceed the provided evidence.

**Audience:**

No

**Audience Explanation:**

The paper lacks the rigorous evidence required by the TMLR community because the theoretical comparisons cannot be drawn in the current state of the paper and the empirical validations lack true generalization metrics.

**Claims And Evidence:**

No

**Claims Explanation:**

Most claims still remain unverified as stated in the Weaknesses above.

**Requested Changes:**

Most changes are already mentioned in the Weaknesses above. Moreover, the authors should use /cite and /citep appropriately in several sections.

---

> ### Author Response · Authors · 2026-06-14
> **Author Response to Reviewer 2X7W**
>
> We thank Reviewer 2X7W for the careful reading of the manuscript and for the constructive comments regarding the positioning of the Brownian formulation, the interpretation of the experiments, and the scope of the theoretical claims.
>
> We agree that the Sobolev and Brownian/Cameron--Martin formulations correspond to different underlying function spaces and therefore should not be interpreted as competing analyses within a fixed hypothesis class. In the revision, we have substantially revised the exposition to make this distinction explicit and have removed statements suggesting universal superiority of the Brownian formulation.
>
> We have also revised the experimental discussion and moderated the conclusions accordingly. The revised manuscript now presents the Brownian/Cameron--Martin framework as an alternative function-space perspective and interprets the experiments as empirical investigations of the corresponding complexity factors rather than as evidence of Brownian superiority.
>
> In addition, we added a dedicated Limitations section discussing the assumptions and scope of the current theory.
>
> A detailed point-by-point response and description of the corresponding revisions are provided in the accompanying response document.
>
> We thank the reviewer again for the helpful feedback.